# The Impacts of Heat Stress on Rumination, Drinking, and Locomotory Behavior, as Registered by Innovative Technologies, and Acid–Base Balance in Fresh Multiparous Dairy Cows

**DOI:** 10.3390/ani14081169

**Published:** 2024-04-13

**Authors:** Ramūnas Antanaitis, Karina Džermeikaitė, Justina Krištolaitytė, Ieva Ribelytė, Agnė Bespalovaitė, Deimantė Bulvičiūtė, Giedrius Palubinskas, Lina Anskienė

**Affiliations:** 1Large Animal Clinic, Veterinary Academy, Lithuanian University of Health Sciences, Tilžės Str. 18, LT-47181 Kaunas, Lithuania; karina.dzermeikaite@lsmu.lt (K.D.); justina.kristolaityte@lsmu.lt (J.K.); ieva.ribelyte@lsmu.lt (I.R.); agne.bespalovaite@stud.lsmu.lt (A.B.); deimante.bulviciute@stud.lsmu.lt (D.B.); 2Department of Animal Breeding, Veterinary Academy, Lithuanian University of Health Sciences, Tilžės Str. 18, LT-47181 Kaunas, Lithuania; giedrius.palubinskas@lsmuni.lt (G.P.); lina.anskiene@lsmu.lt (L.A.)

**Keywords:** thermal stress, acid–base balance, predicted behaviors, dairy cattle

## Abstract

**Simple Summary:**

Heat stress (HS) is a factor that has a negative impact on animal productivity and welfare outcomes, incurring high costs for the dairy sector worldwide. For animals assessed for HS, a reduced feed intake is a conservative response that is most likely an attempt to lower metabolic heat production rates. In the current study, we assess the effects of various heat stress hazards on dairy cows, resulting in considerable decreases in their eating and drinking behavior, reductions in rumination time, and alterations in their locomotion patterns registered by innovative technologies, as well as changes in their acid–base balance.

**Abstract:**

This study hypothesizes that heat stress adversely affects dairy cows, resulting in reduced rumination, altering eating and drinking behaviors, changes in their locomotory patterns, and significant variations in their acid–base balance. The aim of this study was to investigate the impacts of heat stress on rumination, drinking, and locomotory behavior, as registered by innovative technologies, and acid–base balance in fresh multiparous dairy cows. This study was conducted during the summer, from 15 June to 8 July 2023, on a Lithuanian commercial dairy farm. We assessed 350 German Holstein cows that produced an average of 11,400 kg of milk annually throughout their second and subsequent lactation periods. We used the temperature–humidity index (THI) to divide the cows under investigation into three periods: I. high HS—THI >78 (period: 15–23 June 2023); II. medium HS—THI 72–78 (period: 24–30 June 2023); and III. low HS—THI <72 (period: 1–8 July 2023). The appropriate RumiWatch sensor (RWS) parameters were assessed between 15 June 2023 and 8 July 2023. Cows were acclimatized to the rumination, drinking, and locomotory behavior parameters during the adaptation period (1–30 June 2023). The registration process started on 15 June 2023 and terminated on 8 July 2023 and was performed every hour during the 24 h day. The acid–base balance was recorded from 15 June 2023 until 8 July 2023, once per week. The cows’ activity increased by 11.75% in the high HS period compared to the low HS period (*p* < 0.01); high mean differences were detected for rumination, which was 17.67% higher in the high HS period and 13.80% higher in the medium HS period compared to the low HS period (*p* < 0.01); and the change in activity was 12.82% higher in the low HS compared to the medium HS period (*p* < 0.01). Cows under high HS had higher blood urea nitrogen (BUN) levels compared with cows under medium HS (*p* < 0.01). The observed alterations in the rumination, drinking, and locomotory behaviors, in addition to the acid–base balance, highlight the multifaceted impacts of varying heat stress on the physiological and behavioral responses of dairy cows. This suggests that the utilization of advanced technologies may assist dairy farmers in effectively monitoring and controlling heat stress in cows. Additionally, regularly assessing blood urea nitrogen levels can enable farmers to modify their feeding practices, thus promoting optimal cow well-being and productivity.

## 1. Introduction

The progressive and continuous rise in the world’s annual temperature is known as global warming. As a result of the Earth’s temperature rising by roughly 0.18 °C every decade since the early 1980s, it is predicted that by 2100, the planet’s temperature may have increased by 2.1 °C to 3.9 °C [1]. One of the greatest risks to world agriculture has been identified as the combination of excessive summer temperatures and annual temperature rise. Because heat stress (HS) increases the risk of viral and metabolic disorders, lowers milk production, and eventually reduces the fertility of pregnant cows, the dairy industry is especially vulnerable to the effects of climate change [2,3]. Dairy cattle can maintain their body temperature between 38.4 °C and 39.1 °C. At the point when an animal exceeds its thermoneutral zone, which is a surface temperature range of 22–25 °C in moderate climates and 26–37 °C in tropical conditions, it experiences an increase in body heat rather than losing heat. HS occurs when an animal’s core body temperature rises above the normal range [4].

Cows with HS are less able to satisfy their bodies’ energy needs for both milk production and general health, since their energy intake through feed is reduced. This condition results in reductions in milk production rates, reduced walking behavior and milk quality, and increases the animals’ susceptibility to illness [5,6]. The hypothalamic center responsible for appetite is directly negatively impacted by increasing ambient temperatures, leading to a decrease in feed intake behavior. Since ruminating is the primary stimulant for saliva production, heat-stressed cows consume reduced amounts of feed and ruminate less, which reduces the amount of buffering compounds that reach the rumen [7]. Additionally, digestive end products, such as volatile fatty acids (VFAs), are absorbed less effectively because of the redistribution of blood flow to the periphery (in an effort to improve heat dissipation) and the subsequent reduction in blood flow to the gastrointestinal tract. Acute and subclinical rumen acidosis may also result from chronic HS that causes severe or protracted inappetence [8]. 

One potential indicator of HS is the temperature–humidity index (THI) [9]. Studies on heat stress in dairy cows frequently employ the THI, which measures variations in the environment’s temperature and humidity [10]. Recently, livestock nutritionists have frequently used THI, originally used to track human health, to track the connection between crucial THI and the onset of HS in animals [11]. For instance, a decrease in milk output and feed intake occurred when the THI of dairy cattle exceeded 72. A decline in milk supply was noted at a THI of 68 in the US recently due to the discovery that high-producing cows are more susceptible to HS [9].

Therefore, farms need to use effective strategies to identify and minimize HS in dairy cattle. For this purpose, precision livestock farming (PLF) can be used, and effective data management practices can boost production rates in terms of livestock nutrition, animal health, and grazing lot management [12]. At present, PLF technology can be used to monitor the following parameters: temperature, milk yield, movement levels, lying time, and rumination time [13]. A pedometer and noseband sensor have been combined into one system to create the RumiWatch sensor (RWS), a multifunctional gadget with remarkable application, sensitivity, specificity, and use value properties [13]. The RumiWatch noseband sensor has been successfully designed and verified as a scientific monitoring tool for automated evaluations of rumination and eating behavior in dairy cows that are fed in stables. The rumination time has a specificity of 0.98 and a sensitivity of 0.9. The specificity for eating is 0.94, while the sensitivity is 0.84 [14]. While an RWS is more expensive and requires higher maintenance than other precision technologies and necessitates daily maintenance, it can be used to assess grazing management practices and estimate feed consumption rates. The RWS provides high-resolution data from two software packages, which makes it better suited for research purposes than dairy farmers’ everyday assessments [13]. 

Blood acid–base balance analyses are essential to research, according to Gianesella et al. [15], since they are an invaluable tool, particularly when diagnosing acidosis. An altered acidotic response is typically observed in animals with additional clinical illnesses, such as respiratory diseases like pneumonia, according to a study conducted by Gokce et al. [16]. Initially manifesting in a covert manner, metabolic abnormalities are linked to issues concerning the rumen’s fermentation processes. It is clear from the research that the primary step in this process, prior to the production of milk, is nutrition conversion, which is greatly reliant on rumen fermentation [16]. Since the health of dairy cows is closely linked to the functioning of the mammary gland, the overall metabolism level is reflected by the milk constituents [17]. As a result, the molecular markers present in milk provide a clear picture of the metabolic state of dairy cows. Metabolomics has been used in cow studies to assess the likelihood of diseases and to identify biomarkers and pathways associated with metabolic diseases in cows [18]. Based on research by scientists, metabolomics analyzes offer a robust framework for identifying animals and humans who have experienced physiological changes due to exposure to environmental variables. Nevertheless, the metabolic alterations associated with short-term HS in dairy cows during the initial lactation period are still not well understood. HS can have a direct or indirect impact on blood metabolites, as it activates the body’s homeostatic mechanisms to mitigate the harmful effects of HS [5]. Therefore, the components of blood can serve as a measure for evaluating the ability of cattle to adapt to different climates [19]. In addition, heat-stressed cows may display a drop in feed intake and an increase in maintenance needs, which can result in reduced nutritional availability for milk production. Heat stress has been found to disrupt nitrogen metabolism and alter the distribution of nitrogen in dairy cows, resulting in a drop in milk protein content and an increase in milk urea concentration [20]. 

To our knowledge, little is known about the connection between HS and RWS-registered and acid–base balance biomarkers in fresh multiparous dairy cows [21,22,23,24]. Applying technologies like RumiWatch can provide dairy breeders with important information on how to reduce the negative effects of HS on cow health. We hypothesized that heat stress adversely affects dairy cows’ rumination, feeding and drinking behaviors, causes changes in their locomotor parameters registered by innovative technologies and induces significant variations in their acid–base balance. By utilizing advanced technologies to monitor cows’ behaviors and physiological responses, this study seeks to provide valuable insights into the management of heat stress in dairy cattle. To possibly exploit it as a useful tool for animal welfare improvement and observe changes in the bodies of cows during heat stress, farmers and veterinarians could use innovative tools to identify heat-stressed cows. Furthermore, this could lead to more targeted interventions and ultimately improve the overall health and well-being of the animals. By utilizing these tools, farmers and veterinarians can monitor key indicators of heat stress, such as changes in behavior, allowing for timely and effective interventions. Through the implementation of these innovative techniques, farmers can better understand the impact of heat stress on their cows and make informed decisions to ensure their welfare. Ultimately, this proactive approach to identifying and addressing heat stress can help to create a healthier and more comfortable environment for the animals, leading to improved overall productivity and quality of life.

The aim of this study was to investigate the impacts of heat stress on rumination, drinking, and locomotory behavior, as registered by innovative technologies, and acid–base balance in fresh multiparous dairy cows.

## 2. Materials and Methods

### 2.1. Farm, Experiment Design and Duration

The experiment was performed at one Lithuanian dairy farm (coordinates—55.819156, 23.773541) with 850 milking cows. Rubber mat-lined, well-ventilated free-stall buildings were utilized to house the dairy cows. The cows were protected from the elements (sun, precipitation, wind, and dirt) by virtue of their housing in a barn featuring a complete roof and automated ventilation systems that were activated at a temperature of 25 °C. The rubber mats provided comfort and prevented injuries, while the automated ventilation ensured air quality and temperature control. Animals were prevented from going outdoors. All cows were fed a total mixed ration (TMR) balanced according to their nutritional requirements [25,26]. All cows had free access to water. The cows were fed twice a day, at 8:00 and 19:00. The diet was composed of corn silage, alfalfa grass hay, grass silage, sugar beet pulp silage, concentrate mash, and minerals (Table 1). 

This specific diet plan was developed specifically to supply enough nutrition to a 600 kg Holstein cow producing 37 kg of milk every day. The determined chemical compositions are presented in Table 2. 

With the use of a parlor-based milking system, the cows were milked twice daily at approximately 06:00 and 18:00 h. The mean body weight of the cows was 600 kg (±47 kg). The average daily milk yield was 36 kg/d, and the average energy-corrected milk supply per cow was 11,400 kg, with a protein content of 3.6% and a fat content of 4.1%.

This study was performed during the summer period (15 June–8 July 2023) on one Lithuanian dairy farm. The current study involved the examination of 350 Holstein cows who were in their second or subsequent lactation period. These cows were fresh (within the first 60 days postpartum) and had an average annual milk production of 12,000 kg. The temperature–humidity index (THI) was measured on the farm at 10 min intervals using a SmaXtec climate sensor (SmaXtec animal care GmbH, Graz, Austria). The formula used to calculate the temperature–humidity index was THI = 0.8 × T + RH × (T − 14.4) + 46.4 [27]. The heat index was calculated using a heat stress calculator (SmaXtec animal care GmbH, Graz, Austria). We used the THI to divide the cows under investigation into 3 periods [27]: I. high HS—THI > 78 (period: 15–23 June 2023); II. medium HS—THI 72–78 (period: 24–30 June 2023); and III. low HS—THI < 72 (period: 1–8 July 2023). 

### 2.2. Assessments of Rumination, Drinking, and Locomotory Parameters Using RumiWatch Sensors

The RWS tests were performed between 15 June 2023 and 30 July 2023. The cows under investigation acclimatized to the RWS during the adaptation period, which was set from 1 to 15 June 2023. The registration process started on 15 June 2023 and terminated on 8 July 2023, and was performed every hour, 24 h per day. The following RWS parameters were registered [14]: rumination time (min/h)—rumination time during chosen summary interval; eating down time (min/h)—feeding time with head positioned downward during chosen summary interval; eat up time (min/h)—feeding time with head positioned upward during chosen interval; drinking time (min/h)—drinking time during the chosen summary interval; other chewing activities (n/h)—number of jaw movements performed during the chosen summary interval; rumination (n/h)—number of ruminations during the chosen summary interval; eat down chews (n/h)—number of chews performed while the head was positioned downward during the chosen summary interval; eat up chews (n/h)—number of chews performed with head positioned upward during the chosen summary interval; drink gulp (n/h)—number of gulps performed during the chosen summary interval; ruminate boli (n/h)—number of boli regurgitated during the chosen summary interval; ruminations performed per minute (n/min)—number of ruminations performed per minute during the chosen summary interval; walking time (WT) (min/h)—duration of all walking bouts during a predetermined recording period; up time (min/h)—when the pedometer changed its position from a horizontal to a vertical angle for a duration of at least 50 s; down time (min/h)—when the pedometer angle changed its position from a vertical to a horizontal angle for a duration of at least 50 s; activity change (min/h)—activity index (dimensionless), calculated from the variance of the triaxial accelerometer data output; and other activity time (min/h)—other activity time during the chosen summary interval (other activities = activities not classified as any rumination, feeding or drinking activity). 

### 2.3. Registration of Acid–Base Balance

The acid–base balance was recorded from 20 June 2023, until 30 July 2023, once per week. Blood was drawn from each of the 50 cows, which were selected from 350 Holstein cows (in their second or subsequent lactation period), using the jugular venipuncture technique. For this, 1.6 mL heparinized vacutainer blood collection tubes were utilized to assess acid–base balance (Terumo Europe, Leuven, Belgium). The samples were then identified and kept in an ice bath for no more than 30 min until the processing stage. These indices were examined using EPOC acid–base balance analyzers (EPOC, Ottawa, Ontario, Canada). The following parameters were measured: base excess of extracellular fluid (BE ecf), sodium (Na), calcium (C), potassium (K), hematocrit (HCT), chlorides (Cl), hemoglobin concentration (cHgb), bicarbonate (HCO3), hydrogen potential (pH), total carbon dioxide carbon (TCO2), base excess in blood (BE ecf), and lactate (Lac). 

### 2.4. Statistical Analysis

The data analysis was performed using SPSS version 29.0 (IBM Corp., Armonk, New York, NY, USA). The results were expressed as the mean ± standard error of the mean. Descriptive statistics of investigated indicators was carried out according to classes of THI groups. The Spearman’s correlation coefficient was calculated to define the statistical relationships between the evaluated traits and HS group. The general linear model–repeated measures analysis was used for repeated measurements, including, for time periods traits, the same RumiWatch indicator (every hour, 24 h per day) and blood parameters according to the THI groups (observations made from 20 June 2023 to 30 July 2023). The least significant difference (LSD) criterion was used to compare the differences in the mean between group values; a probability value less than 0.05 was considered significant (*p* < 0.05).

## 3. Results

### Descriptive Statistics of the Investigated Indicators

The data analysis of the rumination, drinking, and locomotory behavior parameters presented the highest statistically significant mean differences between the other activity time, other chewing, and activity change parameters during different risk scenarios of HS (Table 3). 

Other activity time was higher by 11.75% in the low HS period compared to the high HS period (*p* < 0.01); high mean differences were detected for other chewing behaviors (17.67% higher in the high HS period and 13.80% higher in the medium HS period compared to the low HS period (*p* < 0.01); and a 12.82% higher activity change was observed in the low HS period compared to the medium HS (*p* < 0.01). 

RMSE—Root Mean Square Error

The mean values for investigated traits assessed every 24 h during different periods of HS (first—high (THI > 78); second—medium (THI 72–78); third—low (THI < 72)) are represented in Figure 1. Data analysis revealed the highest statistically significant differences: rumination time—at 22:00 h (10 p.m.), rumination time was 36.87% lower during the highest HS period (THI1) compared to the medium HS period (THI 2); drinking time—at 17:00 h (5 p.m.), drinking time was 64.38% higher during the highest HS period compared to the medium HS period; eat up chew—at 01:00 h (1 p.m.), eat up chew was 62.33% lower during the medium HS period compared to the lowest HS period; drinking gulp—at 23:00 h (11 p.m.), drinking gulp was 52.49% lower during the highest HS period compared to the medium HS period; chews per bolus—at 17:00 h (5 p.m.), chews per bolus was 66.80% higher during the highest HS period compared to the medium HS period, *p* < 0.01; up time—at 17:00 h (5 p.m.), up time was 68.66% lower during the medium HS period compared to the lowest HS period, *p* < 0.01; chew rumination—at 22:00 h (10 p.m.), chew rumination was 43.14% higher during the highest HS period compared to the medium HS period, *p* < 0.01; activity time—at 17:00 h (5 p.m), activity time was 33.85% higher during the highest HS period compared to the lowest HS; other chew—at 17:00 h (5 p.m.), other chew was 32.35% higher during the highest HS period compared to the lowest HS period; bolus—at 22:00 h (10 p.m.), bolus was 35.60% lower during the highest HS period compared to the medium HS period; chews per minute—at 22:00 h (10 p.m.), chews per minute was 28.87% lower during the highest HS period compared to the medium HS period; activity—at 00:00 h (0 p.m.), activity was 28.97% lower during the highest HS period compared to the medium HS period; down time—at 23:00 h (11 p.m.), down time was 36.19% higher during the highest HS period compared to the lowest HS period, *p* < 0.05.

The data analysis of the investigated parameter relationships shows a weak relationship between HS groups and RWS parameters.

Data analysis revealed a low, negative statistically significant relationship between other activity time, eat up time, drinking time, other chew, eat up chews, chews per bolus, activity change and the THI group, indicating that when the THI increases (THI 1–THI 3), these parameters tend to decrease (*p* < 0.05). Meanwhile, rumination time, eat down time, rumination, eat down chews, gulp, bolus, chews per minute, and activity change showed a low but positive statistically significant relationship with the THI group (*p* < 0.05) (Table 4).

The data analysis of the blood parameters presents the highest statistically significant mean differences for different weeks. We observed high statistically significant mean differences in the blood urea nitrogen (BUN) levels during the week with higher THI (>78) values compared to the week with THI levels of 72–78 (*p* < 0.01). We also observed high statistically significant mean differences in the BUN levels during the other weeks of our study. The BUN level was 19.75% higher in the weeks presenting a THI value of >78 compared to the week presenting a THI value of 72–78, which was 16.89% higher (*p* < 0.01) than the urea level. The urea level was 21.16% higher in the week presenting a THI value of THI > 78 in comparison to the second week and 18.81% higher in comparison to the third week (*p* < 0.01). Other blood parameters remained stable without presenting significant changes (*p* > 0.05) (Table 5). 

The data analysis of the investigated blood parameters’ relationships showed a strong to moderate statistically significant relationship with the THI group.

Data analysis revealed strong, negative statistically significant relationships between THI group and lactate; moderate, negative statistically significant relationships for carbon dioxide pressure and creatinine; and low, negative statistically significant relationships for urea and blood urea nitrogen, indicating that when the THI increases (THI 1–THI 3), these parameters tend to decrease. 

Moderate, positive statistically significant relationships were detected between glucose and hydrogen potential and the THI group (*p* < 0.05) (Figure 2).

## 4. Discussion

Studies detailing dairy cows’ lying and eating habits in relation to housing and management [28], feeding [29], and health [30] have proliferated since the advent of devices that automatically record behavior. In recent years, behavior-recording technologies have also been commercially available, opening up new possibilities for precision cattle husbandry [31]. In the present study, we looked at how heat stress affected fresh multiparous dairy cows’ ruminating, drinking, and locomotory activity as measured by cutting-edge technology and acid–base balance.

According to our findings, when THI was greater than 78, cows exhibited 11.75% more other activity time compared to when THI was less than 72 (*p* < 0.01). Significant mean differences were observed in other chewing behaviors, being 17.67% higher when THI was over 78 and 13.80% higher when THI ranged from 72 to 78, as opposed to when THI was below 72 (*p* < 0.01). Additionally, when the THI was below 72, cows showed an activity change that was 12.82% higher than when the THI ranged from 72 to 78 (*p* < 0.01). The activity level of animals increases with a rise in THI [32]. Elevated THI levels negatively impact animal welfare, leading to altered feeding habits, increased activity levels, and reduced rest periods [7]. Research findings suggest that on days with higher temperatures, the activity levels of bovine animals are lower in the morning and increase in the afternoon [33]. Cattle experiencing heat stress tend to stand for extended periods, which facilitates increased heat dissipation through the skin [7,34]. Additionally, it is possible to hypothesize that dairy cows will experience mental and bodily discomfort as a result of heat stress, which will override any other emotional state, even the strong desire to rest [6]. Adult cattle seldom sleep while standing, so reduced resting time can impact their welfare. Insufficient sleep caused by prolonged standing, particularly a lack of rapid eye movement (REM) sleep, can disrupt the endocrine system, elevate energy consumption, and compromise the immune system [35]. According to sensors, cattle move their heads more while under heat stress. It was discovered that during the summer, cows walk more than they do during the winter [36]. 

In this study, significant mean differences were observed in other chewing behaviors, with a 17.67% increase in the first week (THI > 78) and a 13.80% increase in the second week (THI 72–78) compared to the third week. The variations in rumination behavior under different HS risk categories are consistent with the results determined in previous studies, highlighting the intricate relationship between environmental stressors and animal welfare [23]. Reducing dry matter intake (DMI) is a crucial initial step in alleviating heat stress in dairy cows at air temperatures between 25 and 26 °C, as decreased feed intake is a common conservative response among animals experiencing heat stress [37,38]. The findings of other studies indicate a strong inverse relationship (r = −0.82) between THI and DMI, with an increase in THI of one unit resulting in a daily reduction in DMI of 0.45 kg [39]. Furthermore, by controlling the metabolic rate and heat production, a change in DMI served as a protective mechanism to preserve thermal balance [40]. When dairy cows were exposed to cold conditions (THI = 38.9) as opposed to hot conditions (THI = 73.9), Hill and Wall [41] found that there was an 11.5% increase in DMI. In contrast, our findings were corroborated by the finding that DMI decreased by 0.51 kg for each unit rise in THI between 73 and 83 units [38]. Specifically, inconsistent DMI responses to THI conditions and milk output are probably the result of disturbed energy metabolism under ongoing cold and wet conditions [42]. The hypothalamic cooling center signals the medial satiety center to affect the appetite center, which results in hypophagia, or a decrease in feed intake levels [43]. Consequently, the mammary gland cannot produce a lot of milk due to a lack of nutrients [38]. Bouraoui et al. [44] discovered that cows under heat stress conditions had a 9.6% lower DMI. In comparison to cows in the thermoneutral zone, Spiers et al. (2004) [45] observed that cows exposed to heat stress displayed a daily DMI reduction of 14.6 kg/cow and a daily milk yield reduction of 11.8 kg/cow. Further investigations are necessary to comprehend the potential impact of decreased DMI levels during heat stress conditions on the physiological and behavioral reactions of dry dairy cows [45].

Additionally, the fact that the other chewing activity factor increased at the same time during the first and second weeks compared to the third week supports the idea that changes in feeding patterns help the body to adapt to heat stress [46,47]. The high R-squared values obtained in our study (ranging from 0.5323 to 0.8374) suggest a strong predictive relationship between the duration of HS exposure and the corresponding changes in eating, eat chewing, and down times. These results are consistent with St-Pierre et al.’s [48] theory of cumulative heat load influencing behavioral patterns over time. It is noteworthy that the observed increase in activity change during low HS risk conditions in the third week was not a straightforward reversal of the patterns observed under high-risk conditions. According to Bohmanova et al. [49], animals subjected to prolonged heat stress may develop adaptive mechanisms that result in altered behavior, even in less stressful circumstances, which can explain this nuanced response. 

The results of the study reveal significant changes in various behavioral parameters over a 24 h period, providing valuable insights into the diurnal variations in the activities of the subjects under study. The observed changes in eating time, ranging from 4.21 to 13.61, reflect the substantial variations in the duration of feeding activities conducted throughout the day. This aligns with the results determined in previous studies on ruminant feeding behavior, where animals typically exhibited periodic bouts of eating interspersed with periods of rest or other activities [50]. The observed increase in eating time suggests potential adjustments to feeding patterns, possibly influenced by factors such as food availability, palatability, or nutritional requirements. Similarly, the alterations in drinking time from 0.20 to 0.52 highlight the variations in water intake levels over a 24 h period. Changes in water consumption levels can be influenced by other factors such as ambient temperature, dietary composition, and the overall hydration status of the animals [51]. These changes can be attributed to various factors, such as social interactions, environmental stimuli, and physiological needs. The observed increase in the rumination time, in particular, is consistent with the natural digestive processes in ruminants, where cud-chewing plays a vital role in the breakdown of fibrous feed materials [52]. Additionally, the variations in bolus and down time indicate changes in resting and regurgitation behaviors. Down time, which ranged from 16.48 to 24.95, reflects the time spent lying down and resting. 

The data analysis of the blood parameters revealed notable variations in the mean values across different HS risks, with the most substantially significant differences observed for BUN levels. This means that when the THI is higher than 78, BUN levels are statistically significantly higher by 21.66% compared to the second week, when the THI is between 72 and 78 (*p* < 0.01). These results underscore the sensitivity of BUN and urea indicators to variations in thermal stress levels. Cattle experiencing HS undergo behavioral and metabolic alterations meant to preserve homeothermy, frequently at the cost of lower output and profitability values [38]. For instance, in HS cows, Bernabucci and Calamari (2010) [10] observed a reduction in rumen microbial (MCP) synthesis behavior. On the other hand, other studies have shown that the increased water consumption level caused by HS may hasten the rate of microbial passage [53] and improve microbial development [54], which, in turn, may lead to the synthesis of MCP. The lack of variations in MCP production between treatments in the current study likely occurred because similar amounts of dietary protein were consumed. Furthermore, decreased splanchnic blood flow can potentially restrict the absorption of amino acids (AA) [55]; for instance, we noted that portal flow tended to decline in HS conditions, but that it was more closely associated with the DMI level than the surrounding temperature [56]. The fact that plasma AA levels decreased and BUN levels increased, along with similar MCP production levels, suggested that extra-mammary AA use increased in HS conditions. Thus, the decrease in milk protein levels observed in HS conditions could be the result of a lack of available precursors supplied to the mammary gland. It appears that blood AA utilization is redirected away from milk protein synthesis during HS [56]. 

The findings from our analysis reveal intriguing patterns in physiological responses to variations within the Temperature–Humidity Index (THI) groups, providing valuable insights into animal adaptation mechanisms under different environmental stress levels. The significant negative correlation between the THI group and lactate suggests that as animals are exposed to higher-THI conditions (from THI 1 to THI 3), their lactate production decreases. This could be interpreted as an adaptive physiological response where animals may enhance their heat tolerance or stress resilience over time, leading to a more efficient metabolic process that produces less lactate, a common indicator of stress and anaerobic metabolism [57]. Moreover, the moderate negative correlation observed with carbon dioxide pressure and creatinine further supports the notion that a higher THI, which indicates increased heat stress, is associated with altered metabolic and respiratory functions. These changes might reflect compensatory mechanisms to maintain homeostasis, such as adjusted ventilation rates to manage body temperature and modified kidney functions to ensure electrolyte balance and prevent dehydration [10]. The slight negative correlation with urea and blood urea nitrogen (BUN) underlines a potentially more subtle adjustment in protein metabolism or nitrogen excretion processes as the THI increases. This could imply that animals under higher thermal stress might optimize their protein utilization or nitrogenous waste management to mitigate the effects of heat stress on their overall metabolic efficiency [54]. These findings underscore the complexity of physiological responses to environmental stressors like temperature and humidity. Understanding these responses is crucial for developing strategies to improve animal welfare and productivity in the face of climate variability. Future research should aim to explore the underlying mechanisms of these correlations further, assess the long-term impacts of varying THI conditions on animal health and performance, and evaluate the effectiveness of different mitigation strategies to enhance resilience to thermal stress [45]. 

While there are variations in thermal tolerance among different livestock species, with ruminants generally having higher tolerance than monogastric animals, there is also significant variability in the tolerance levels among breeds within each species. Therefore, there may be a chance to enhance animals’ ability to withstand high temperatures by utilizing genetic techniques [58]. In Northeastern Australia, the Belmont Red hybrid breed was specifically bred to enhance cow output by improving heat tolerance, resistance to parasites, and their ability to withstand periods of severe under-nutrition. Modelling results indicate that enhancing heat stress resistance through breeding is likely to have a detrimental impact on livestock productivity [59]. An alternative approach to gene editing involves the introduction of the heat tolerance gene (SLICK), which is naturally present in heat-resistant cattle breeds like Senepol and Brahman, into the genome of breeds like Holstein that are adapted to more moderate climates. This gene enhances the cattle’s ability to regulate their body temperature, thereby decreasing the likelihood of heat stress [60]. Regardless of their location and breeding, contemporary dairy cows suffer from the negative impacts of HS, ultimately resulting in decreased fertility. The detrimental impacts of HS begin at the early stages of oocyte development and continue throughout later stages, affecting its ability to be fertilized. HS also has unfavorable effects on the estrus cycle, estrus behavior, embryo development, implantation, uterine environment, and even extends to the fetal calf. HS can induce acyclicity in cows. An immediate solution to decrease fertility declines in dairy cows at the farm is to properly diagnose HS and implement efficient cooling measures, regardless of the animals’ stage of life. Additional measures to mitigate the reduction in fertility during HS include providing proper nutrition and care, limiting the occurrence of diseases and mastitis, utilizing semen from bulls that have been cooled, implementing timed artificial inseminations (AI), employing hormonal therapies, and utilizing embryo transfer technology [8]. An optimal and enduring solution would involve carefully strategized breeding programs aimed at enhancing fertility and developing tolerance to heat stress [61]. Additionally, ongoing research into new technologies and management practices for heat stress mitigation will continue to improve the overall resilience of livestock in hot climates. Ultimately, a comprehensive approach that combines genetic improvement, management strategies, and technological advancements will be key to ensuring the long-term fertility and productivity of animals in the face of increasing heat stress challenges.

This study presented some limitations, including its single-farm focus on Holstein cows during a specific period, limiting its generalizability. Also, the analysis of the investigated parameter relationships reveals a weak correlation between rumination, drinking, and locomotor behavior, and acid–base balance, with the statistically significant correlation coefficients ranging from 0.323 to 0.375. This is likely due to the small sample size of animals used in the study. Factors such as individual variability, external influences on behavior, and the absence of long-term impact assessments highlight the need for further research to refine heat stress management strategies in dairy farming practices. Despite these limitations, this study provided valuable insights into the dynamic nature of ruminant behavior and blood parameters resulting from heat stress, laying the groundwork for future research on this area and improved management practices.

## 5. Conclusions

Based on our study’s findings, we advise dairy farmers to employ innovative technologies for the real-time monitoring and management of heat stress in cows. During periods of high THI (>78), it is crucial to closely monitor changes in activity time and chewing behaviors, as significant increases in these behaviors are indicative of elevated heat stress levels. Farmers should also adjust their nutritional plans based on the continuous monitoring of blood urea nitrogen levels. This innovative approach, leveraging advanced technologies, is vital for maintaining the health and productivity of dairy cows under various climatic conditions.

## Figures and Tables

**Figure 1 animals-14-01169-f001:**
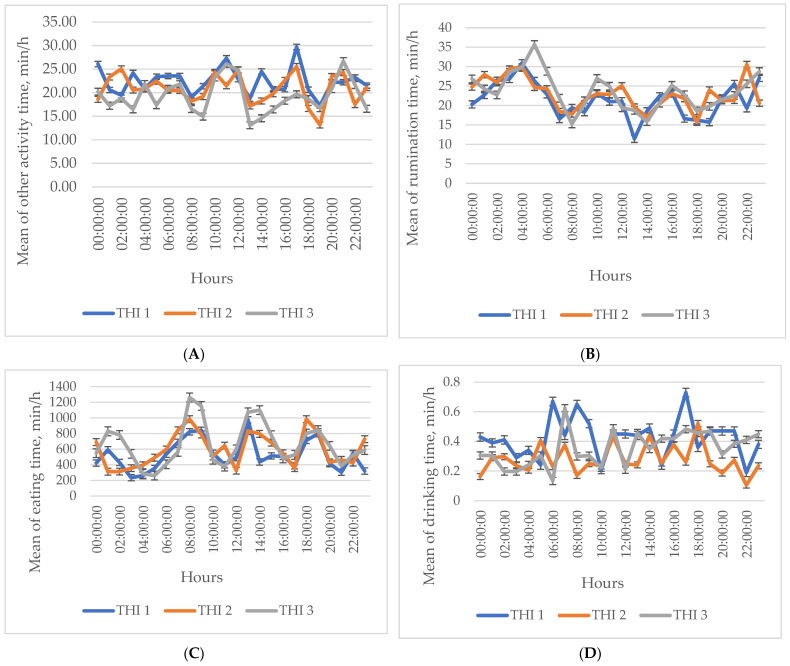
The mean and standard error of the changes in rumination, drinking, and locomotory behavior parameters depending on the time of day. (**A**) Activity time; (**B**) rumination time; (**C**) eating time; (**D**) drinking time; (**E**) other chew; (**F**) chew rumination; (**G**) eat chew; (**H**) drinking gulp; (**I**) bolus; (**J**) chews per minute; (**K**) chews per bolus; (**L**) activity; (**M**) uptime; (**N**) down time. Three periods of temperature humidity index: THI 1, THI 2, THI 3.

**Figure 2 animals-14-01169-f002:**
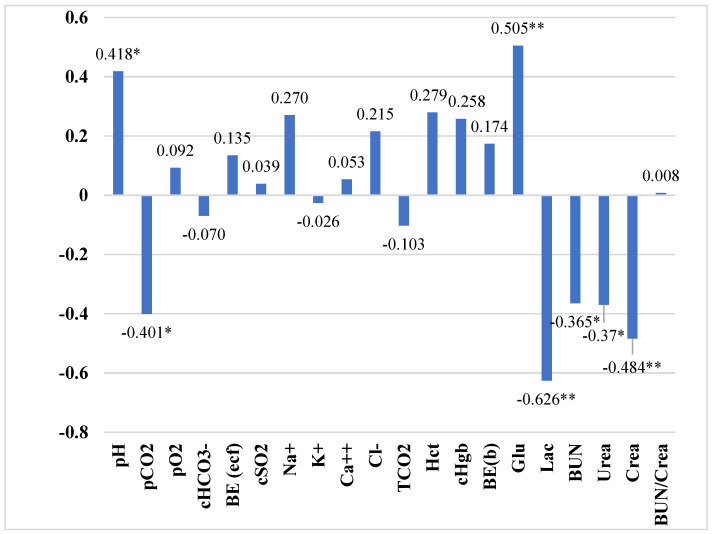
Spearman’s correlation coefficient between the THI group and blood parameters. Correlation is significant. pH—hydrogen potential; pCO2—partial carbon dioxide pressure; pO2—partial oxygen pressure; HCO3—bicarbonate; BE (ecf)—base excess in the extracellular fluid; sO2—oxygen saturation; Na+—sodium; K+—potassium; Ca++—ionized calcium; Cl-chlorides; TCO2—total carbon dioxide in the blood; Hct—hematocrit; cHgb—hemoglobin concentration; BE—base excess in the blood; Glu—glucose; Lac—lactate; BUN—blood urea nitrogen; Crea—creatinine; BUN/Crea—blood urea nitrogen and creatinine ratio. Correlation is significant. * *p* < 0.05, ** *p* < 0.001.

**Table 1 animals-14-01169-t001:** Ingredients of the total mixed ration.

Ingredient	%
Corn silage	25
Alfalfa grass hay	5
Grass silage	20
Sugar beet pulp silage	15
Concentrate mash	30
Mineral mixture	5

**Table 2 animals-14-01169-t002:** Chemical composition of the feeding ration.

Parameter	Composition
Dry matter (DM)	48.8%
Neutral detergent fiber	28.2% of DM
Acid detergent fiber	19.8% of DM
Non-fiber carbs	38.7% of DM
Crude protein	15.8% of DM
Net energy for lactation	MJ/kg DM

**Table 3 animals-14-01169-t003:** The changes in data regarding rumination, drinking, and locomotory behavior across different risk scenarios of heat stress.

Parameters	High (THI > 78) ^a^	Medium (THI 72–78) ^b^	Low (THI < 72) ^c^	RMSE
Other activity time (min/h)	22.80 ^c^	21.83	20.12 ^a^	0.71
Rumination time (min/h)	21.31	22.25	22.80	0.77
Eat down time (min/h)	7.20 ^c^	8.18	8.69 ^a^	0.49
Eat up time (min/h)	8.32	7.49	8.08	0.42
Drink time (min/h)	0.39 ^b^	0.27 ^a^	0.32	0.35
Other chew (n/h)	133.28 ^c^	127.30 ^c^	109.73 ^a,b^	0.49
Rumination (n/h)	1422.42 ^b^	1569.72 ^a^	1548.79	54.10
Eat down chews (n/h)	544.02	605.99	647.20	38.06
Eat up chews (n/h)	576.64	526.95	558.34	31.20
Gulp (n/h)	286.11	344.49	316.33	21.23
Bolus (n/h)	23.48	24.19	25.02	0.87
Chews per minute (n/min)	63.96	66.30	65.72	1.52
Chews per bolus (n/boli)	5.59 ^b^	3.91 ^a^	4.93	0.57
Walking time (WT) (min/h)	85.83 ^b^	96.65 ^a,c^	86.38 ^b^	2.80
Up time (min/h)	10.64	10.04	10.27	0.89
Down time (min/h)	21.02	20.58	20.43	1.06
Activity change (min/h)	7.80	7.07	8.11	0.24

The letters a, b, and c indicate the statistically significant differences between THI groups; *p* < 0.05.

**Table 4 animals-14-01169-t004:** Spearman’s correlation coefficient between THI group and RWS parameters.

	OAT	RT	EDT	EUT	DT	OC	CR	EDC	EUC	G	B	CPM	CPB	WT	UT	DT	AC
THI group	−0.083 *	0.054 **	0.069 *	−0.033	−0.040 *	−0.103 *	0.063 *	0.072 *	−0.028	0.041 *	0.047 *	0.039 *	−0.039 *	0.000	−0.014	0.006	−0.037 *

Correlation is significant. * *p* < 0.05, ** *p* < 0.001. OAT—other activity time; RT—rumination time; EDT—eat down time; EUT—eat up time; DT—drink time; OC—other chew; CR—chew rumination; EDC—eat down chews; EUC—eat up chews; G—gulp; B—bolus; CPM—chews per minute; CPB—chews per bolus; WT—walking time; UT—up time; DT—down time; AC—activity change.

**Table 5 animals-14-01169-t005:** Descriptive statistics of the investigated blood parameters (mean ± standard error of mean).

Blood Parameters	Risk of HS			
	High (THI > 78) ^a^	Medium (THI 72–78) ^b^	Low (THI < 72) ^c^	RMSE
pH	7.41 ± 0.017	7.46 ± 0.019	7.47 ± 0.013	0.23
pCO2	46.77 ± 1.850	41.17 ± 2.033	40.57 ± 1.773	2.67
pO2	191.93 ± 9.683	153.83 ± 16.603	202.35 ± 6.586	19.35
HCO3-	29.52 ± 0.551	28.67 ± 0.541	29.21 ± 0.679	0.84
BE (ecf)	4.88 ± 0.644	4.74 ± 0.623	5.50 ± 0.584	0.87
cSO2	99.59 ± 0.074	98.00 ± 0.851	99.75 ± 0.037	0.70
Na+	135.30 ± 0.539	135.90 ± 0.690	136.50 ± 0.500	0.82
K+	4.17 ± 0.086	4.02 ± 0.137	4.15 ± 0.069	0.14
Ca++	1.18 ± 0.020	1.17 ± 0.025	1.19 ± 0.014	0.03
Cl−	100.70 ± 0.539	101.40 ± 0.748	101.70 ± 0.539	0.87
TCO2	29.410 ± 0.546	28.450 ± 0.540	28.950 ± 0.688	0.84
Hct	25.30 ± 0.423	25.90 ± 0.657	26.40 ± 0.427	0.73
cHgb	8.66 ± 0.128	8.80 ± 0.230	8.98 ± 0.150	0.23
BE(b)	4.32 ± 0.591	4.350 ± 0.584	5.05 ± 0.504	0.79
Glu	3.01 ± 0.145 ^c^	3.12 ± 0.081 ^c^	3.45 ± 0.054 ^a,b^	0.14
Lac	2.03 ± 0.150 ^c^	1.45 ± 0.131	1.27 ± 0.086 ^a^	0.17
BUN	14.80 ± 0.814 ^b,c^	12.30 ± 0.700 ^a^	12.60 ± 0.670 ^a^	1.03
Urea	5.28 ± 0.287 ^b,c^	4.36 ± 0.249 ^a^	4.49 ± 0.230 ^a^	0.36
Crea	81.90 ± 1.616	77.60 ± 2.798	70.40 ± 3.661	3.99
BUN/Crea	15.96 ± 0.729	14.27 ± 1.288	16.02 ± 0.841	1.39

The letters a, b, and c indicate statistically significant differences between HS groups; *p* < 0.05. pH—hydrogen potential; pCO2—partial carbon dioxide pressure; pO2—partial oxygen pressure; HCO3—bicarbonate; BE (ecf)—base excess in the extracellular fluid; sO2—oxygen saturation; Na+—sodium; K+—potassium; Ca++—ionized calcium; Cl-chlorides; TCO2—total carbon dioxide in the blood; Hct—hematocrit; cHgb—hemoglobin concentration; BE—base excess in the blood; Glu—glucose; Lac—lactate; BUN—blood urea nitrogen; Crea—creatinine; BUN/Crea—blood urea nitrogen and creatinine ratio. RMSE—root mean square error.

## Data Availability

The data provided in this study can be found in the publication.

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
