# Peer review of "The Impacts of Heat Stress on Rumination, Drinking, and Locomotory Behavior, as Registered by Innovative Technologies, and Acid–Base Balance in Fresh Multiparous Dairy Cows"

_animals, 2024, doi:10.3390/ani14081169_

Round 1

Reviewer 1 Report (Previous Reviewer 2)

Comments and Suggestions for Authors

General comments:

The manuscript animals-2913415, entitled " The Impacts of Heat Stress on the Factors of Rumination, Drinking, and Locomotory Behavior in Dairy Cows, Registered by Innovative Technologies, and Blood Gas Parameters In Fresh Multiparous Dairy Cows" with Mr. Antanaitis as first author was mostly well revised by the authors. The reviewer comments were considered to a large extent in the revision and the authors left only some of my questions unanswered. However, some minor inconsistencies and logical mistakes can still be found in the manuscript.  Various comments for suggested changes to the text could be find in the following section.

 Detailed comments:

Line 10            The hyperlink of author D.B. is not complete. Please add the full hyperlink.

Line 19            Please replace the term “movement” by the term “locomotory”

Line 20            The last two sentences are very similar and should therefore be combined.

Line 35            23.06.2023 is still included in the first two survey periods in this paragraph. Please correct the sentence.

Line 42            Please harmonize within the manuscript “percent” or “%”!

Line 50            Please emphasize the practical benefits for farms at the end of the abstract.

Line 51            The full stop at the end of the keywords is not necessary.

Line 64            This sentence is too long. Please split into two sentences. The information “like India” is not necessary here.

Line 88            In my opinion there is missing a verb in this sentence.

Line 92            Please replace the term “identified” by “identify”.

Line 93            Please rephrase: For this purpose precision livestock farming (PLF) can be used…

Line 98            Please rephrase: …have been combined into one system to create the RumiWatch sensor…

Line 104          The potential connection between heat stress and blood gas analyses is not clear from the explanations. Please describe your hypothesis and cite appropriate references.

Line 107          Please delete the second full stop at the end of the sentence.

Line 120          Please replace the term “eating” by “feeding”.

Line 124          The objective of the study still remains unclear. Please specify the objectives of the study and emphasize the novelty value. What practical benefits do you expect for dairy farmers?

Line 187          Please insert a blanket space between “Ottawa).” and the new sentence…

Line 206          Please harmonize the number of blanket spaces used in the brackets (THI >78) instead of (THI> 78).

Line 235          Please harmonize within the manuscript “p<0.05” or “p < 0.05”!

Table 1           Tables should be self-explanatory, so the table heading should also include relevant information about the statistical analysis. Use the same number of decimal places in the table. I would sort the rows according to the individual behaviors (rumination, eating, activity, etc.). This makes it easier to compare the results with each other.

Line 222          The space between “drinking time” and “was detected” is in my opinion too large. Maybe there are two blanket spaces?

Figure 1          The units on the y-axis are missing (hours etc.). Why did you choose a mixture of lines and bars in the figures? I would use three different colored lines here to make the illustration easier to understand. Perhaps the scaling could be adjusted to make it easier to identify differences. Please correct: Mean of chews per minute. Figures should be self-explanatory, so abbreviations should be written out in full or integrated into footnotes.

Line 255          Please add a blanket space between “THI” and “3”.

Table 2           The table is not easy to read. Maybe you can use abbreviations to describe the behaviours to shorten the table (abbreviations should be described in the footnotes); Correlations are usually calculated with metric values. Why are you comparing the RWS parameters with the ordinal THI groups here? Please explain the background of the statistical analyses and give reasons for their selection.

Table 3           Please widen the table to make the contents easier to read. a,b,c should be formatted as superscripts. The spacing and the indents used should be standardized in the footnotes. The abbreviation RMSE is missing.

Figure 2          The figure does not seem to be complete. To which blood parameter does 0.008 belong? Correlations are usually calculated with metric values. Why are you comparing the blood parameters with the ordinal THI groups here? Please explain the background of the statistical analyses and give reasons for their selection. Please add the abbreviations used in this figure in the footnotes.

Line 294          Please add a blanket space between “THI” and “3”.

Line 314          Please add the month in the bracket (2023.07.01-08.07)?

Line 328          Please delete the second full stop after the sentence.

Line 332          Why do you cite a study, which deals with rats? Are there no suitable studies on the effects of heat stress on dairy cows?

Line 345          Please insert a blanket space between “heat stress” and the brackets…

Line 359          What do you mean with thermos comfort zone (thermoneutral zone)?

Line 366          Please check the references in the text. Sometimes the years are inserted and sometimes not.

Line 368          Where can the values be found in the manuscript? Please show the numbers in the results or delete this sentence.

Line 455          Please check the specifications for the references again. In several cases, the abbreviated journals have been inserted and not their full names (e.g. 458, 460, 531, 543)

Line 474          The name of the first author is not correct.

Line 497          Maybe “Animal” instead of “animal”?

Line 493          The journal is missing

Line 570          Please delete “1” after the title.

Line 502          The name of the first author is perhaps not correct.

Line 574          The names of the authors are not correct.

Author Response

Dear Editor,

Authors appreciate the comments that assist in enhancing the article. All suggested modifications have been incorporated into the manuscript and are marked in yellow and track changes.

Sincerely,

Ramūnas Antanaitis

Comments and Suggestions for Authors

Authors comments, corrections and answers

General comments:

The manuscript animals-2913415, entitled " The Impacts of Heat Stress on the Factors of Rumination, Drinking, and Locomotory Behavior in Dairy Cows, Registered by Innovative Technologies, and Blood Gas Parameters In Fresh Multiparous Dairy Cows" with Mr. Antanaitis as first author was mostly well revised by the authors. The reviewer comments were considered to a large extent in the revision and the authors left only some of my questions unanswered. However, some minor inconsistencies and logical mistakes can still be found in the manuscript.  Various comments for suggested changes to the text could be find in the following section.

Thank you for the remarks.

 Detailed comments:

Line 10         The hyperlink of author D.B. is not complete. Please add the full hyperlink.

Corrected

Line 19         Please replace the term “movement” by the term “locomotory”

We change the term "movement" to "locomotory".

Line 20         The last two sentences are very similar and should therefore be combined.

We combined these two sentences to " In the current study, we assess the effects of various heat stress hazards on dairy cows, resulting in considerable decreases in their eating and drinking behavior, reduction in rumination time, and alterations in their locomotion patterns registered by innovative technologies as well as changes in their blood gas parameters."

Line 35         23.06.2023 is still included in the first two survey periods in this paragraph. Please correct the sentence.

We corrected the sentence by changing the second period to "2023.06.24 – 06.30".

Line 42         Please harmonize within the manuscript “percent” or “%”!

We harmonized manuscript with "%".

Line 50         Please emphasize the practical benefits for farms at the end of the abstract.

We added sentences: "It suggests that the utilisation of advanced technologies may assist dairy farmers effectively monitor and control heat stress in cows. Additionally, regularly assessing blood urea nitrogen levels can enable farmers to modify feeding practices, thus promoting optimal cow well-being and productivity."

Line 51         The full stop at the end of the keywords is not necessary.

We removed the full stop at the end of keywords.

Line 64         This sentence is too long. Please split into two sentences. The information “like India” is not necessary here.

We corrected this sentence into: “At the point when an animal exceeds its thermoneutral zone, which is a surface temperature range of 22–25 °C in moderate climates and 26–37 °C in tropical conditions, it experiences an increase in body heat rather than losing heat. HS occurs when an animal's core body temperature rises above the normal range.”

Line 88         In my opinion there is missing a verb in this sentence.

We corrected this sentence into: “For instance, a decrease in milk output and feed intake occurred when the THI of dairy cattle exceeded 72”.

Line 92         Please replace the term “identified” by “identify”.

We replaced the term “identified” into “identify”.

Line 93         Please rephrase: For this purpose precision livestock farming (PLF) can be used…

We rephrased “For this purpose precision livestock farming (PLF) can be used, ...”

Line 98         Please rephrase: …have been combined into one system to create the RumiWatch sensor…

We rephrased “…have been combined into one system to create the RumiWatch sensor..."

Line 104       The potential connection between heat stress and blood gas analyses is not clear from the explanations. Please describe your hypothesis and cite appropriate references.

We added more information: “Metabolomics has been used in cow studies to assess the likelihood of diseases and to identify biomarkers and pathways associated with metabolic diseases in cows. Based on research by scientists, metabolomics analyzes offer a robust framework for identifying animals and humans who have experienced physiological changes due to exposure to environmental variables. Nevertheless, the metabolic alterations associated with short-term HS in dairy cows during the initial lactation period are still not well understood. HS can have a direct or indirect impact on blood metabolites, as it activates the body's homeostatic mechanisms to mitigate the harmful effects of HS. Therefore, the components of blood can serve as a measure for evaluating the ability of cattle to adapt to different climates. In addition, heat-stressed cows may have a drop in feed intake and an increase in maintenance need, which can result in reduced nutritional availability for milk production. Heat stress has been found to disrupt nitrogen metabolism and alter the distribution of nitrogen in dairy cows, resulting in a drop in milk protein content and an increase in milk urea concentration “.

Line 107       Please delete the second full stop at the end of the sentence.

We deleted the second full stop at the end of the sentence.

Line 120       Please replace the term “eating” by “feeding”.

We replaced the term “eating” into “feeding”.

Line 124       The objective of the study still remains unclear. Please specify the objectives of the study and emphasize the novelty value. What practical benefits do you expect for dairy farmers?

We added information: “To possibly exploit it as a useful tool for animal welfare improvement and observe changes in the bodies of cows during heat stress, farmers and veterinarians could use innovative tools to identify heat-stressed cows. Furthermore, this could lead to more targeted interventions and ultimately improve the overall health and well-being of the animals. By utilizing these tools, farmers and veterinarians can monitor key indicators of heat stress, such as changes in behavior, allowing for timely and effective intervention. Through the implementation of these innovative techniques, farmers can better understand the impact of heat stress on their cows and make informed decisions to ensure their welfare. Ultimately, this proactive approach to identifying and addressing heat stress can help to create a healthier and more comfortable environment for the animals, leading to improved overall productivity and quality of life.”

Line 187       Please insert a blanket space between “Ottawa).” and the new sentence…

We inserted a blanked space between “Ottawa).” and the new sentence.

Line 206       Please harmonize the number of blanket spaces used in the brackets (THI >78) instead of (THI> 78).

We harmonized the number of blanket spaces used in the brackets (THI >78) instead of (THI> 78).

Line 235       Please harmonize within the manuscript “p<0.05” or “p < 0.05”!

We harmonized manuscript with “P<0.05".

Table 1         Tables should be self-explanatory, so the table heading should also include relevant information about the statistical analysis. Use the same number of decimal places in the table. I would sort the rows according to the individual behaviors (rumination, eating, activity, etc.). This makes it easier to compare the results with each other.

The letters a, b, and c indicate the statistically significant differences between THI groups, not between indicators.

RMSE decimal places – is corrected

Line 222       The space between “drinking time” and “was detected” is in my opinion too large. Maybe there are two blanket spaces?

We removed a blanket space.

Figure 1        The units on the y-axis are missing (hours etc.). Why did you choose a mixture of lines and bars in the figures? I would use three different colored lines here to make the illustration easier to understand. Perhaps the scaling could be adjusted to make it easier to identify differences. Please correct: Mean of chews per minute. Figures should be self-explanatory, so abbreviations should be written out in full or integrated into footnotes.

Corrected: units and design of a figure 1.

Line 255       Please add a blanket space between “THI” and “3”.

We added a blanket space.

Table 2         The table is not easy to read. Maybe you can use abbreviations to describe the behaviours to shorten the table (abbreviations should be described in the footnotes); Correlations are usually calculated with metric values. Why are you comparing the RWS parameters with the ordinal THI groups here? Please explain the background of the statistical analyses and give reasons for their selection.

Abbreviations – are added in a Table 2.

We don’t have THI values for each day / hour, we have general information during which period was high, or medium, or low THI.

We used the THI to divide the cows under investigation into 3 groups [21]: I. high HS—THI > 78 (period: 2023.06.15- 06.23); II. medium HS—THI 72-78 (period: 2023.06.24 – 06.30); and III. low HS—THI < 72 (period: 2023.07.01- 08).

Table 3         Please widen the table to make the contents easier to read. a,b,c should be formatted as superscripts. The spacing and the indents used should be standardized in the footnotes. The abbreviation RMSE is missing.

Corrected

Figure 2        The figure does not seem to be complete. To which blood parameter does 0.008 belong? Correlations are usually calculated with metric values. Why are you comparing the blood parameters with the ordinal THI groups here? Please explain the background of the statistical analyses and give reasons for their selection. Please add the abbreviations used in this figure in the footnotes.

The figure was completed, but somehow BUN/CREA trait was deleted, we added it (r=0.008).

We don’t have THI values for each day / hour, we have general information during which period was high, or medium, or low THI.

We used the THI to divide the cows under investigation into 3 groups [21]: I. high HS—THI > 78 (period: 2023.06.15- 06.23); II. medium HS—THI 72-78 (period: 2023.06.24 – 06.30); and III. low HS—THI < 72 (period: 2023.07.01- 08).

Basically, that is why we used Spearman correlation.

Line 294       Please add a blanket space between “THI” and “3”.

We added a blanket space.

Line 314       Please add the month in the bracket (2023.07.01-08.07)?

We added the month in the bracket (2023.07.01-07.08).

Line 328       Please delete the second full stop after the sentence.

We deleted the second full stop at the end of the sentence.

Line 332       Why do you cite a study, which deals with rats? Are there no suitable studies on the effects of heat stress on dairy cows?

We deleted the sentence and added more information: " Additionaly, it is possible to hypothesise that dairy cows will experience mental and bodily discomfort as a result of heat stress, which will override any other emotional state, even the strong desire to rest. Adult cattle seldom sleep while standing, so reduced resting time can impact their welfare. Insufficient sleep caused by prolonged standing, particularly a lack of rapid eye movement (REM) sleep, can disrupt the endocrine system, elevate energy consumption, and compromise the immune system."

Line 345       Please insert a blanket space between “heat stress” and the brackets…

We added a blanket space.

Line 359       What do you mean with thermos comfort zone (thermoneutral zone)?

We corrected the term into "thermoneutral zone".

Line 366       Please check the references in the text. Sometimes the years are inserted and sometimes not.

We inserted missing years.

Line 368       Where can the values be found in the manuscript? Please show the numbers in the results or delete this sentence.

We deleted this sentence

Line 455       Please check the specifications for the references again. In several cases, the abbreviated journals have been inserted and not their full names (e.g. 458, 460, 531, 543)

Corrected

Line 474       The name of the first author is not correct.

Corrected to – “Ajeet, K.; Meena, K. Effect of Heat Stress in Tropical Livestock and Different Strategies for Its Amelioration. Journal of Stress Physiology & Biochemistry 2011, 7, 45–54.”

Line 497       Maybe “Animal” instead of “animal”?

Corrected

Line 493       The journal is missing

We added - Polish journal of veterinary sciences 2021, 24, 253–260, doi:10.24425/pjvs.2021.137660.

Line 570       Please delete “1” after the title.

We deleted “1” after the title.

Line 502       The name of the first author is perhaps not correct.

The name of the first author is M. J. VandeHaar, we wrote it correctly.

Line 574       The names of the authors are not correct.

Corrected to – “Paterson, J. A.; Belyea, R. l.; Bowman, J. P.; Kerley, M. S.; Williams, J. E. The Impact of Forage Quality and Supplementation Regimen on Ruminant Animal Intake and Performance. In Forage Quality, Evaluation, and Utilization; John Wiley & Sons, Ltd, 1994; pp. 59–114 ISBN 978-0-89118-579-6.”

Reviewer 2 Report (Previous Reviewer 3)

Comments and Suggestions for Authors

The manuscript focuses on the effects of heat stress in cows. The topic is very interesting, but the article is confusingly written, both in its material and methods and in its results. 

It should be better explained how heat stress periods are defined and how statistical results are performed. For example, how the correlations were performed, how interesting the numbers in Figure 1 are, why the RMSE is so large. In addition, unnecessary information is given, such as the number of cows on the farm, which is repeated twice and adds nothing.

In my opinion, the authors should focus on the objectives, provide the results that can satisfy the working hypothesis and discuss them.

Author Response

Dear Editor,

Authors appreciate the comments that assist in enhancing the article. All suggested modifications have been incorporated into the manuscript and are marked in yellow and track changes.

Sincerely,

Ramūnas Antanaitis

Comments: The manuscript focuses on the effects of heat stress in cows. The topic is very interesting, but the article is confusingly written, both in its material and methods and in its results. 

Answer: we corrected whole manuscript according yours and others reviewers recommendations.

Comments: It should be better explained how heat stress periods are defined and how statistical results are performed. For example, how the correlations were performed, how interesting the numbers in Figure 1 are, why the RMSE is so large. In addition, unnecessary information is given, such as the number of cows on the farm, which is repeated twice and adds nothing.

Answer: We added information  about heat stress periods. “During the first week, we observed a high risk of heat stress (THI> 78). In the second week, the risk level was moderate (THI 72–78). In the third week, the risk level decreased to low (THI <72).The formula used to calculate the temperature humidity index was THI = 0.8 × T + RH × (T - 14.4) + 46.4 [27]. The heat index was calculated using a heat stress calculator (SmaXtec animal care GmbH, Graz, Austria). We used the THI to divide the cows un-der investigation into 3 groups [27]: I. high HS—THI > 78 (period: 2023.06.15- 06.23); II. medium HS—THI 72-78 (period: 2023.06.24 – 06.30); and III. low HS—THI < 72 (pe-riod: 2023.07.01- 08)”.

The results are expressed as the mean ± standard error of the mean. Descriptive statistics of investigated indicators was carried out according to classes of THI groups. The Spearman's correlation coefficient was calculated to define the statistical relationships between the evaluated traits and HS group. The general linear model–Repeated measures analysis was used for repeated measurements, including for time periods traits the same RumiWatch indicator (every hour, 24 hours per day), blood parameters according to the THI groups (observations made from 20 June 2023 to 30 July 2023). The LSD criterion was used to compare the differences in the mean between group values; a probability value less than 0.05 was considered significant (P < 0.05).

We corrected figure 1. We used three different coloured lines to make the figure easier to understand.

RMSE – was recalculated to RMSE between groups (one stand. Error for all 3 THI groups).

Comments:In my opinion, the authors should focus on the objectives, provide the results that can satisfy the working hypothesis and discuss them.

Answer: we corrected introduction part to – “To possibly exploit it as a useful tool for animal welfare improvement and observe changes in the bodies of cows during heat stress, farmers and veterinarians could use innovative tools to identify heat-stressed cows. Furthermore, this could lead to more targeted interventions and ultimately improve the overall health and well-being of the animals. By utilizing these tools, farmers and veterinarians can monitor key indicators of heat stress, such as changes in behavior, allowing for timely and effective intervention. Through the implementation of these innovative techniques, farmers can better understand the impact of heat stress on their cows and make informed decisions to ensure their welfare. Ultimately, this proactive approach to identifying and addressing heat stress can help to create a healthier and more comfortable environment for the animals, leading to improved overall productivity and quality of life.

The aim of this study was to investigate the impacts of heat stress on rumination, drinking, and locomotory behavior, registered by innovative technologies, and acid-base balance in fresh multiparous dairy cows”

Reviewer 3 Report (New Reviewer)

Comments and Suggestions for Authors

Here is the review of animals- 2913415.

Title: Needs formatting. The color is different from black.

Simple summary: Needs formatting. The color is different from black.

Abstract: Needs formatting. The color is different from black. Keep checking through the paper.

L20-25: Repeated twice. Line 20 to 23 and line 25 to 27.

L30: You do not need to report the number of cows in the entire farm. You must report only how many cows you used in the study. Was it a commercial farm?

L31-33: Suggest revising to …We assessed 350 German Holstein cows, projected 305-d milk production (x ± sd) and parity (x ± sd)”.

L37-39: Suggest revising to …” Cows were acclimatized to the rumination, drinking, and locomotory behavior parameters during the adaptation period (from 1 to 14 June 2023)” …

L39-40: I suggest you use the same date format throughout the paper. Sometimes you use from 1 to 14 June 2023 and sometimes you use 2023.06.23 – 06.30.

L39-40: There is a reason of why the registration process was performed every hour during the 24-hour day instead of every 5 minutes?

L42: Suggest revising to …” Cows’ activity increased by 11.75 percent in high HS compared to the low HS (P< 0.01);” … You can report <0.01 do not need <0.001. Keep the same format along the paper. Instead of reporting results using percentage, I would like to see LSMEANS per treatment for all parameters presented in the paper.  

L46: I Suggest revising to …” Cows in high HS had higher blood urea nitrogen (BUN) compared with cows in medium HS (P< 0.001) … I recommend that you report the LSMEANS for both treatments. The way you wrote the sentence was very confusing. Keep this format along the paper.

Keywords: Suggest using key words that are not in the title.

L59: Please add a citation to support your statement. Paper [1] does not support the following. The progressive and continuous rise in the world's annual temperature is known as 56 global warming. As a result of the Earth's temperature rising by roughly 0.18 °C every 57 decade since the early 1980s, it is predicted that by 2100, the planet's temperature may 58 have increased by 2.1 °C to 3.9 °C

L64-68: I Suggest you divide the following sentence in two, it’s too long and it gets confusing. “At the point when an animal transgresses its thermoneutral zone, which is a surface temperature of 22–25 °C in mild and 26–37 °C in tropical climate environments, like India, this results in an increase in body heat as opposed to heat loss, where the animal’s core body temperature increases to temperatures higher than the normal range, resulting in HS [4]”.

L70: Suggest revising to …” feed” … instead of food.

L76: Include citation.

L80: Suggest revising to …” This increases the total VFA content in the rumen and decrease pH levels” …

L82: include citation.

L92: Suggest revising to …” Therefore, farms need to use effective strategies to identify” …

L93-95: Suggest revising to …” For this purpose, it can be used the precision livestock farming (PLF) and effective data management practices that aim to boost production rates in terms of livestock nutrition, animal health, and grazing lot management [11].

L99-100: Suggest revising to …” While an RWS is more expensive and requires higher maintenance

than other precision technologies, it can be” ….

L107: Suggest revising to …” Gokce et al. [14].” …

L110: What do you mean by nutrition conversion?

L113: Suggest revising to …” milk constituents [15].” … Make sure to correct the grammar in the paper.  

L115-117: Suggest revising to …” To our knowledge, little is known about the connection between HS and RWS-registered and blood gas biomarkers in fresh multiparous dairy cows” …

L119-122: Suggest revising to …” We hypothesized that heat stress adversely affects dairy cows rumination, eating and drinking behaviors, cause changes in their locomotor parameters and cause significant variations in their blood gas parameters” …

 L128-129: I am confused with the following. All 850 cows kept in free-stall barns were fed a total mixed ration (TMR) balanced according to their nutritional requirements. You used 350 cows as mentioned in the abstract or 850 cows?

L128-129: “All 850 cows kept in free-stall barns were fed a total mixed ration (TMR) balanced according to their nutritional requirements”. You should cite (NASEM, 2021) instead.

L130-131: Suggest revising to …”All cows had free access to Water” …

L131-132: Suggest revising to …”The cows were fed twice a day, at 800 and 1900h,” … I recommend you deleting “as per the multiparous, high-yielding cows' standard feeding schedule:”

L132: Suggest revising to …” The diet was composed by corn silage, alfalfa grass hay, grass silage, sugar beet pulp silage, concentrate mash, and minerals” …
You do not have to report the percentage of each ingredient in the diet here. You should have that in table 1.

L136-139: L: Report the dietary chemical composition in the table.

L139-140: Suggest revising to” Cows were milked at approximately 0600 and 1800 h”.

L141: Include average daily milk yield.

L141: Delete “Over the course of a single lactation period

L145-148: Suggest rephrasing the sentence, it’s confusing. You should include DIM and SD per treatment, lactation order average per treatment and SD.

L148: Temperature and humidity were recorded in which frequency? every 5 min? Please describe it.

L159: Make sure the following is in the correct format. “The registration process started on 20 June 2023” …

L193: Include more details on how the data were analyzed. It’s not clear.

L203-205: I suggest you rephrase the following: it’s confusing the way you report the results. Try to keep it simple, it will be easier to read and understand the results. “The data analysis of the rumination, drinking, and locomotory behavior parameters presented the highest statistically significant mean differences between the other activity time, other chewing, activity change parameters, during different risk scenarios of HS (Table 1)” I previously mentioned an example on how to report the results.

L206-207: Again, it’s hard to understand what you want to say. Be more direct. We observed that, in the first week, the risk of HS was high (THI> 78); in the second it was medium (THI 72–78); and in the third it was low (THI <72).

Table 1: Why do you have small letters in the treatments? High (THI>78)a. What that means?

Table 1: Please revise the letters you used to show difference between treatments.

Table 1: What you have to say about the RMSE - Root Mean Square Error reported? Are those numbers correct?

Table 1: Please follow journal format.

L219-220: I do not think this is the most appropriate way to report your results. I gave you one example of how to report your results. “Data analysis revealed the highest statistically significant differences” …

Figure 1: Format figure as required by the journal. Should have Figure 1A, 1B, etc?

Figure 1: data for THI 1 is not shown as THI 2 and THI 3. Can you please provide an explanation for this?

Table 3: Include p-value.

L292-295: Very hard to understand the main point. I suggest you rephrase the following. The results need to be in a simple way to help readers to understand it. Data analysis revealed strong, negative statistically significant relationship between THI group and lactate; moderate - carbon dioxide pressure, creatinine; low –urea, and blood urea nitrogen, indicating that when THI group gets higher (THI 1 – THI3), these parameters tend to decrease (P<0.05).

Discussion: Most be improved, the first paragraph does not discuss any results, second paragraph repeats the results. 

L342: Include citation.

Comments on the Quality of English Language

Extensive editing of English language required.

Author Response

Dear Reviewer,

Authors appreciate the comments that assist in enhancing the article. All suggested modifications have been incorporated into the manuscript and are marked in yellow and track changes.

Sincerely,

Ramūnas Antanaitis

Comments and Suggestions for Authors

Authors comments, corrections, and answers

Here is the review of animals- 2913415.

Title: Needs formatting. The color is different from black.

We have changed the title colour to black.

Simple summary: Needs formatting. The color is different from black.

We have changed the simple summary colour to black.

Abstract: Needs formatting. The color is different from black. Keep checking through the paper.

We have changed the abstract colour to black.

L20-25: Repeated twice. Line 20 to 23 and line 25 to 27.

We corrected line 20 to 23 to "In the current study, we assess the effects of various heat stress hazards on dairy cows, resulting in considerable decreases in their eating and drinking behavior, reduction in rumination time, and alterations in their locomotion patterns registered by innovative technologies as well as changes in their blood gas parameters."

L30: You do not need to report the number of cows in the entire farm. You must report only how many cows you used in the study. Was it a commercial farm?

We deleted the number of cows in the entire farm and added information that it was commercial farm.

L31-33: Suggest revising to … “We assessed 350 German Holstein cows, projected 305-d milk production (x ± sd) and parity (x ± sd)”.

“We assessed 350 German Holstein cows, projected 305-d milk production (1140 ± sd) and parity (2 ± 1)”

L37-39: Suggest revising to …” Cows were acclimatized to the rumination, drinking, and locomotory behavior parameters during the adaptation period (from 1 to 14 June 2023)” …

We corrected the sentence: "Cows were acclimatized to the rumination, drinking, and locomotory behavior parameters during the adaptation period (2023.06.01 - 2023.06.30)."

L39-40: I suggest you use the same date format throughout the paper. Sometimes you use from 1 to 14 June 2023 and sometimes you use 2023.06.23 – 06.30.

We corrected the date format throughout the paper.

L39-40: There is a reason of why the registration process was performed every hour during the 24-hour day instead of every 5 minutes?

The activity's output comprises unprocessed classification summaries with a resolution of 1 minute. By utilising the software's additional conversion and classification capabilities, it is possible to generate consolidated summaries of animal activity, such as those with a one-hour resolution. As a result, the recorded sensor data are subjected to further validity tests as part of the analysis algorithm, which serve to prevent the interpretation of measured values as invalid or flawed.

L42: Suggest revising to …” Cows’ activity increased by 11.75 percent in high HS compared to the low HS (P< 0.01);” … You can report <0.01 do not need <0.001. Keep the same format along the paper. Instead of reporting results using percentage, I would like to see LSMEANS per treatment for all parameters presented in the paper. 

We corrected the P value to P<0.01 instead of P<0.001.

According to our experience, we have used percentages to present the results in the articles, and this approach has been effective.

L46: I Suggest revising to …” Cows in high HS had higher blood urea nitrogen (BUN) compared with cows in medium HS (P< 0.001)” … I recommend that you report the LSMEANS for both treatments. The way you wrote the sentence was very confusing. Keep this format along the paper.

We corrected the sentence to: “Cows in high HS had higher blood urea nitrogen (BUN) compared with cows in medium HS (P<0.001)”.

Keywords: Suggest using key words that are not in the title.

We changed keywords to: “thermal stress; acid-base balance; predicted behaviors; dairy cattle“.

L59: Please add a citation to support your statement. Paper [1] does not support the following. The progressive and continuous rise in the world's annual temperature is known as 56 global warming. As a result of the Earth's temperature rising by roughly 0.18 °C every 57 decade since the early 1980s, it is predicted that by 2100, the planet's temperature may 58 have increased by 2.1 °C to 3.9 °C

We added citations to support our statment.

L64-68: I Suggest you divide the following sentence in two, it’s too long and it gets confusing. “At the point when an animal transgresses its thermoneutral zone, which is a surface temperature of 22–25 °C in mild and 26–37 °C in tropical climate environments, like India, this results in an increase in body heat as opposed to heat loss, where the animal’s core body temperature increases to temperatures higher than the normal range, resulting in HS [4]”.

We corrected this sentence into: “At the point when an animal exceeds its thermoneutral zone, which is a surface temperature range of 22–25 °C in moderate climates and 26–37 °C in tropical conditions, it experiences an increase in body heat rather than losing heat. HS occurs when an animal's core body temperature rises above the normal range.”

L70: Suggest revising to …” feed” … instead of food.

We revised “food” into “feed”.

L76: Include citation.

We included citations.

L80: Suggest revising to …” This increases the total VFA content in the rumen and decrease pH levels” …

We corrected sentence to: “This increases the total VFA content in the rumen and decrease pH levels”.

L82: include citation.

We included citations.

L92: Suggest revising to …” Therefore, farms need to use effective strategies to identify” …

We corrected the sentence to: “Therefore, farms need to use effective strategies to identify...”

L93-95: Suggest revising to …” For this purpose, it can be used the precision livestock farming (PLF) and effective data management practices that aim to boost production rates in terms of livestock nutrition, animal health, and grazing lot management [11].”

We corrected the sentence to: “For this purpose, precision livestock farming (PLF) can be used, and effective data management practices can boost production rates in terms of livestock nutrition, animal health, and grazing lot management."

L99-100: Suggest revising to …” While an RWS is more expensive and requires higher maintenance

than other precision technologies, it can be” ….

We corrected the sentence to: “While an RWS is more expensive and requires higher maintenance than other precision technologies, it can be...”.

L107: Suggest revising to …” Gokce et al. [14].” …

We corrected the sentence to: “... Gokce et al. [14].”

L110: What do you mean by nutrition conversion?

L113: Suggest revising to …” milk constituents [15].” … Make sure to correct the grammar in the paper. 

We corrected the sentence to: “... milk constituents [15].”

L115-117: Suggest revising to …” To our knowledge, little is known about the connection between HS and RWS-registered and blood gas biomarkers in fresh multiparous dairy cows” …

We corrected the sentence to: “To our knowledge, little is known about the connection between HS and RWS-registered and blood gas biomarkers in fresh multiparous dairy cows.”

L119-122: Suggest revising to …” We hypothesized that heat stress adversely affects dairy cows rumination, eating and drinking behaviors, cause changes in their locomotor parameters and cause significant variations in their blood gas parameters” …

We corrected the sentence to: "We hypothesized that heat stress adversely affects dairy cows rumination, feeding and drinking behaviors, cause changes in their locomotor parameters registered by innovative technologies and induce significant variations in their blood gas parameters."

L128-129: I am confused with the following. All 850 cows kept in free-stall barns were fed a total mixed ration (TMR) balanced according to their nutritional requirements. You used 350 cows as mentioned in the abstract or 850 cows?

We updated the text to reflect the fact that the farm had 850 cows, but we only examined 350 of them.

 L128-129: “All 850 cows kept in free-stall barns were fed a total mixed ration (TMR) balanced according to their nutritional requirements”. You should cite (NASEM, 2021) instead.

We corrected citation.

L130-131: Suggest revising to …”All cows had free access to Water” …

We corrected the sentence to: “All cows had free access to water.”

L131-132: Suggest revising to …”The cows were fed twice a day, at 800 and 1900h,” … I recommend you deleting “as per the multiparous, high-yielding cows' standard feeding schedule:”

We removed “as per the multiparous, high-yielding cows' standard feeding schedule”.

L132: Suggest revising to …” The diet was composed by corn silage, alfalfa grass hay, grass silage, sugar beet pulp silage, concentrate mash, and minerals” …
You do not have to report the percentage of each ingredient in the diet here. You should have that in table 1.

We corrected to - “The diet was composed by corn silage, alfalfa grass hay, grass silage, sugar beet pulp silage, concentrate mash, and minerals (Table 1).

Table 1. Ingredients of the total mixed ration.

L136-139: L: Report the dietary chemical composition in the table.

We corrected to - “The determined chemical compositions are presented in Table 2.

Table 2. Chemical composition of the feeding ration.

L139-140: Suggest revising to” Cows were milked at approximately 0600 and 1800 h”.

We corrected the sentence to: " With the use of a parlor-based milking system, the cows were milked twice daily at approximately 0600 and 1800 h."

L141: Include average daily milk yield.

We corrected the sentence and included an average daily milk yield: “The  average daily milk yield was 36kg/d., and the average energy-corrected milk supply per cow was 11,400 kg, with a protein content of 3.6% and a fat content of 4.1%."

L141: Delete “Over the course of a single lactation period”

We deleted “Over the course of a single lactation period”.

L145-148: Suggest rephrasing the sentence, it’s confusing. You should include DIM and SD per treatment, lactation order average per treatment and SD.

We rephrased the sentence to: “The current study involved the examination of 350 Holstein cows who were in their second or subsequent lactation period. These cows were fresh (within the first 60 days post partum) and had an average annual milk production of 12,000 kg."

L148: Temperature and humidity were recorded in which frequency? every 5 min? Please describe it.

We corrected the sentence to: “The temperature humidity index (THI) was measured on the farm at 10 min intervals using a SmaXtec climate sensor (SmaXtec animal care GmbH, Graz, Austria)."

L159: Make sure the following is in the correct format. “The registration process started on 20 June 2023” …

We corrected the sentence to: “The registration process started on 2023.06.20 and terminated on 2023.07.30, and was performed every hour, 24 hours per day."

L193: Include more details on how the data were analyzed. It’s not clear.

Corrected

2.4. Statistical analysis

The data analysis was performed using SPSS version 29.0 (IBM Corp., Armonk, New York, NY, USA). The results are expressed as the mean ± standard error of the mean. Descriptive statistics of investigated indicators was carried out according to classes of THI groups. The Spearman's correlation coefficient was calculated to define the statistical relationships between the evaluated traits. The general linear model–Repeated measures analysis was used for repeated measurements, including for time periods traits the same RumiWatch indicator (every hour, 24 hours per day), blood parameters according to the THI groups (observations made from 20 June 2023 to 30 July 2023). The LSD criterion was used to compare the differences in the mean between group values; a probability value less than 0.05 was considered significant (P < 0.05).

L203-205: I suggest you rephrase the following: it’s confusing the way you report the results. Try to keep it simple, it will be easier to read and understand the results. “The data analysis of the rumination, drinking, and locomotory behavior parameters presented the highest statistically significant mean differences between the other activity time, other chewing, activity change parameters, during different risk scenarios of HS (Table 1)” I previously mentioned an example on how to report the results.

Accepted.

L206-207: Again, it’s hard to understand what you want to say. Be more direct. We observed that, in the first week, the risk of HS was high (THI> 78); in the second it was medium (THI 72–78); and in the third it was low (THI <72).

We corrected the sentence to: "During the first week, we observed a high risk of heat stress (THI> 78). In the second week, the risk level was moderate (THI 72–78). In the third week, the risk level decreased to low (THI <72)."

Table 1: Why do you have small letters in the treatments? High (THI>78)a. What that means?

The explanation is mentioned at the bottom of the Table 1: “The letters a, b, and c indicate the statistically significant differences between weeks; P < 0.05."

Table 1: Please revise the letters you used to show difference between treatments.

The letters a, b, and c indicate the statistically significant differences between groups of THI; P < 0.05.

High THI  – a; medium THI – b, low THI – c.

Table 1: What you have to say about the RMSE - Root Mean Square Error reported? Are those numbers correct?

At the beginning in the manuscript was used a regression analysis with Y – equation and R – squared results, but according to one of the reviewers we have changed it to RMSE.

RMSE – was recalculated to RMSE between groups (one stand. Error for all 3 THI groups).

Table 1: Please follow journal format.

Corrected

L219-220: I do not think this is the most appropriate way to report your results. I gave you one example of how to report your results. “Data analysis revealed the highest statistically significant differences” …

Accepted / corrected

Figure 1: Format figure as required by the journal. Should have Figure 1A, 1B, etc?

Corrected

Figure 1: data for THI 1 is not shown as THI 2 and THI 3. Can you please provide an explanation for this?

We presented the means of the studied indicators at different study periods according to THI groups: THI1 , THI2, THI 3 ( not THI measured values of each hour).

Table 3: Include p-value.

It was included before. We found a statistically significant mean differences just in: Glu—glucose; Lac—lactate; BUN

L292-295: Very hard to understand the main point. I suggest you rephrase the following. The results need to be in a simple way to help readers to understand it. Data analysis revealed strong, negative statistically significant relationship between THI group and lactate; moderate - carbon dioxide pressure, creatinine; low –urea, and blood urea nitrogen, indicating that when THI group gets higher (THI 1 – THI3), these parameters tend to decrease (P<0.05).

We corrected the sentence to: “The data analysis revealed a strong negative correlation between the THI group and lactate (P<0.05). There was a moderate negative correlation with carbon dioxide pressure and creatinine, and a weak negative correlation with urea and blood urea nitrogen. This indicates that when the THI group increases (THI 1 - THI3), these parameters tend to decrease (P<0.05)”.”

Discussion: Most be improved, the first paragraph does not discuss any results, second paragraph repeats the results. 

We deleted this information

L342: Include citation.

We included a citation.

Reviewer 4 Report (New Reviewer)

Comments and Suggestions for Authors

 Review, paper no. animals-2913415 entitle The Impacts of Heat Stress on the Factors of Rumination, Drinking, and Locomotory Behavior in Dairy Cows, Registered by Innovative Technologies, and Blood Gas Parameters In Fresh Multiparous Dairy Cows. The authors have used the standard journal format in manuscript writing. There are few minor observations that the authors should address before final submission.

Specific comments:

Use acid base-balance instead of "blood gas parameters" throughout the manuscript. In some places leave the previous term.

Abstract: 

Is sufficiently presented (methods, results, general conclusions).

Introduction: The introduction section is sufficient and analytically and adequately covers the need for the study.

Line 80. VFAs do not have strong ruminal pH lowering properties. Correctly identify the impact of HS.

Line 87. livestock nutritionists ?

Line 99. Indicate sensitivity and specificity.

Methods: The methodology is sufficiently presented. However, it has a few inaccuracies.

The methodology lacks information on keeping cows.

Line. 151. The SmaXtec system does not calculate THI according to the formula THI = 0.8 × D + RH × (T 14.4) + 46.4. Correct.

Line 183. Specify the number of cows.

Line. 187. Base excess in the ex- tracellular fluid (BE ecf) ? Correct.

What statistical test was used to compare the means.

Result and discussion

The results of the study are analytically presented. Figures are adequate explain the findings of the study.

The results in Table 1 do not indicate heat stress (THI >78). Please explain.

Please provide recommendations for breeding practice (specific actions to mitigate heat).

Add detailed discussion on HS and acid base-balance. The studies probably compensated for blood gas parameters. Interpreted parameters changes. In acute heat stress, the changes are different.

Conclusion: In conclusion, generalizations are given.

Author Response

Dear Reviewer,

Authors appreciate the comments that assist in enhancing the article. All suggested modifications have been incorporated into the manuscript and are marked in yellow and track changes.

Sincerely,

Ramūnas Antanaitis

Comments and Suggestions for Authors

Authors comments, corrections, and answers

Review, paper no. animals-2913415 entitle „The Impacts of Heat Stress on the Factors of Rumination, Drinking, and Locomotory Behavior in Dairy Cows, Registered by Innovative Technologies, and Blood Gas Parameters In Fresh Multiparous Dairy Cows”. The authors have used the standard journal format in manuscript writing. There are few minor observations that the authors should address before final submission.

Thank you for the remarks.

Specific comments:

Use acid base-balance instead of "blood gas parameters" throughout the manuscript. In some places leave the previous term.

We corrected with “acid-base balance” in whole manuscript

Abstract: 

Is sufficiently presented (methods, results, general conclusions).

Thank you for the remarks.

Introduction: The introduction section is sufficient and analytically and adequately covers the need for the study.

Thank you for the remarks.

Line 80. VFAs do not have strong ruminal pH lowering properties. Correctly identify the impact of HS.

We deleted this sentence

Line 87. livestock nutritionists?

We have changed the sentence for clarity to: “Recently, livestock nutritionists had frequently used THI, originally used to track human health, to track the connection between crucial THI and the onset of HS in animals.”

Line 99. Indicate sensitivity and specificity.

We added information: “The RumiWatch noseband sensor has been successfully designed and verified as a scientific monitoring tool for automated evaluations of rumination and eating behavior in dairy cows that are fed in stables. The rumination time has a specificity of 0.98 and a sensitivity of 0.9. The specificity for eating is 0.94, while the sensitivity is 0.84“.

Methods: The methodology is sufficiently presented. However, it has a few inaccuracies.

Thank you for the remarks.

The methodology lacks information on keeping cows.

We added information about keeping cows: “The experiment was performed at one Lithuanian dairy farm with 850 milking cows. Rubber mat-lined, well-ventilated free-stall buildings were utilized to house the dairy cows. The cows were protected from the elements (sun, precipitation, wind, and dirt) by virtue of their housing in a barn featuring a complete roof and automated ventilation systems that engaged at a temperature of 25 °C. The rubber mats provided comfort and prevented injuries, while the automated ventilation ensured air quality and temperature control. It was prohibited for animals to enter an outdoor sanctuary “.

Line. 151. The SmaXtec system does not calculate THI according to the formula THI = 0.8 × D + RH × (T 14.4) + 46.4. Correct.

We corrected the formula to: “THI = 0.8 × T + RH × (T - 14.4) + 46.4"

Line 183. Specify the number of cows.

We corrected to – “Blood was drawn from each of the 50 cows using the jugular venipuncture technique. For this, 1.6 mL heparinized vacutainer blood collection tubes were utilized to assess acid-base balance ...“

Line. 187. Base excess in the ex- tracellular fluid (BE ecf) ? Correct.

We corrected to - “base excess of extracellular fluid (BE ecf)”

What statistical test was used to compare the means.

The general linear model–Repeated measures analysis was used for repeated measurements, including for time periods traits the same RumiWatch indicator (every hour, 24 hours per day), blood parameters according to the THI groups (observations made from 20 June 2023 to 30 July 2023). The LSD criterion was used to compare the differences in the mean between group values; a probability value less than 0.05 was considered significant (P < 0.05).

Result and discussion

The results of the study are analytically presented. Figures are adequate explain the findings of the study.

Thank you for the remarks.

The results in Table 1 do not indicate heat stress (THI >78). Please explain.

We added information - “During the first week, we observed a high risk of heat stress (THI> 78). In the second week, the risk level was moderate (THI 72–78). In the third week, the risk level decreased to low (THI <72).”

Please provide recommendations for breeding practice (specific actions to mitigate heat).

We added information about breeding practice: “While there are variations in thermal tolerance among different livestock species, with ruminants generally having higher tolerance than monogastric, there are also significant variability in tolerance levels among breeds within each species. Therefore, there may be a chance to enhance the animals' ability to withstand high temperatures by utilizing genetic techniques. In North Eastern Australia, the Belmont Red hybrid breed was specifically bred to enhance cow output by improving heat tolerance, resistance to parasites, and ability to withstand periods of severe under-nutrition. Modelling results indicate that enhancing heat stress resistance through breeding is likely to have a detrimental impact on livestock productivity. An alternative approach to gene editing involves the introduction of the heat tolerance gene (SLICK), which is naturally present in heat-resistant cattle breeds like Senepol and Brahman, into the genome of breeds like Holstein that are adapted to more moderate climates. This gene enhances the cattle's ability to regulate their body temperature, thereby decreasing the likelihood of heat stress. Regardless of their location and breeding, contemporary dairy cows suffer from the negative impacts of HS, which ultimately result in decreased fertility. The detrimental impacts of HS begin at the early stages of oocyte development and continue throughout later stages, affecting its ability to be fertilized. HS also has unfavorable effects on the oestrus cycle, oestrus behavior, embryo development, implantation, uterine environment, and even extends to the foetal calf. HS can induce acyclicity in cows. An immediate solution to decrease fertility declines in dairy cows at the farm is to properly diagnose HS and implement efficient cooling measures, regardless of the animals' stage of life. Additional measures to mitigate the reduction in fertility during HS include providing proper nutrition and care, limiting the occurrence of diseases and mastitis, utilizing semen from bulls that have been cooled, implementing timed artificial inseminations (AI), employing hormonal therapies, and utilizing embryo transfer technology. An optimal and enduring solution would involve carefully strategized breeding programmes aimed at enhancing fertility and developing tolerance to heat stress. Additionally, ongoing research into new technologies and management practices for heat stress mitigation will continue to improve the overall resilience of livestock in hot climates. Ultimately, a comprehensive approach that combines genetic improvement, management strategies, and technological advancements will be key to ensuring the long-term fertility and productivity of animals in the face of increasing heat stress challenges.”

Add detailed discussion on HS and acid base-balance. The studies probably compensated for blood gas parameters. Interpreted parameters changes. In acute heat stress, the changes are different.

We added information in a disscusion section -

The findings from our analysis reveal intriguing patterns in physiological responses to variations within the Temperature-Humidity Index (THI) groups, providing valuable insights into animal adaptation mechanisms under different environmental stress levels. The significant negative correlation between THI group and lactate suggests that as animals are exposed to higher THI conditions (from THI 1 to THI 3), their lactate production decreases. This could be interpreted as an adaptive physiological response where animals may enhance their heat tolerance or stress resilience over time, leading to a more efficient metabolic process that produces less lactate, a common indicator of stress and anaerobic metabolism [57]. Moreover, the moderate negative correlation observed with carbon dioxide pressure and creatinine further supports the notion that higher THI groups, which indicate increased heat stress, are associated with altered metabolic and respiratory functions. These changes might reflect compensatory mechanisms to maintain homeostasis, such as adjusted ventilation rates to manage body temperature and modified kidney functions to ensure electrolyte balance and prevent dehydration [10]. The slight negative correlation with urea and blood urea nitrogen (BUN) underlines a potentially more subtle adjustment in protein metabolism or nitrogen excretion processes as the THI group increases. This could imply that animals under higher thermal stress might optimize their protein utilization or nitrogenous waste management to mitigate the effects of heat stress on their overall metabolic efficiency [54]. These findings underscore the complexity of physiological responses to environmental stressors like temperature and humidity. Understanding these responses is crucial for developing strategies to improve animal welfare and productivity in the face of climate variability. Future research should aim to explore the underlying mechanisms of these correlations further, assess the long-term impacts of varying THI conditions on animal health and performance, and evaluate the effectiveness of different mitigation strategies to enhance resilience to thermal stress [45].

Conclusion: In conclusion, generalizations are given.

Thank you for the remarks.

Round 2

Reviewer 1 Report (Previous Reviewer 2)

Comments and Suggestions for Authors

General comments:

 The manuscript animals-2913415, entitled "The Impacts of Heat Stress on Rumination, Drinking, and Locomotory Behavior, Registered by Innovative Technologies, and Acid Base - Balance In Fresh Multiparous Dairy Cows" with Mr. Antanaitis as first author was mostly well revised by the authors. The reviewer comments were considered in the revision and the authors left none of my questions unanswered. However, some minor inconsistencies and logical mistakes can still be found in the manuscript. It is also annoying that some of the recurring errors were not corrected in the manuscript despite being pointed out several times. Various comments for suggested changes to the text could be find in the following section.

 Detailed comments:

Line 8              Some of the hyperlinks are not correct (e.g. „.lt“ is missing). Please correct

Line 21            The term „acid base-balance“ is written in a different font size.

Line 33            Please add a full stop between 2023.06.15 and „-„ and between „-„ and 06.23)

Line 52            Please delete the blanket space after „acid-base balance“

Line 52            Please delete the blanket space before „predicted behaviors“

Line 61            Please delete one blanket space between „.“ and „At the point“

Line 87            There is missing a full stop at the end of the sentence.

Line 107          Please add a blanket space between „assessments“ and [13].

Line 107          What does „OBJ“ mean?

Line 112          Please harmonize within the manuscript: References with year numbers or not?

Line 140          Please delete one blanket space between „cattle.“ and „To possibly“

Line 158          Please add the coordinates.

Line 178          Please harmonize within the manuscript: 0600 or 06:00?

Line 179          Please delete the full stop after 36 kg/d.

Line 180          Please harmonize within the manuscript blanket space between value and unit (%) or not?

Line 182          Please delete one blanket space between „farm.“ and „The current study“

Line 220          Please specify: Which 50 cows?

Line 239          The abbreviation LSD was not introduced before

Line 241          Please harmonize within the manuscript: P<0.05 or P < 0.05?

Line 247          Please add a blanket space between THI and >78

Line 249          Please add a blanket space between < and 72

Table 3           Tables should be self-explanatory, so the table heading should also include relevant information about the statistical analysis.

Line 259          Please add a blanket space between THI and >78

Line 259          Please add a blanket space between < and 72

Figure 1          Please use the same font as in the text

                        Sometimes the legend is not correct (THI 1) instead of (TH I1)

                        F is not formatted like the others (hours)

                        Mean of chews per minute

Table 4           footnotes should be formatted similar: e.g. Rumination time – RT

Table 5           THI >78 instead of THI>78

                        footnotes should be formatted similar: e.g. pH – hydrogen potential

Line 345          Please add a blanket space between „-„ and „urea“

Line 346          Please add a blanket space between „THI“ and „3“

Figure 2          footnotes should be formatted similar: e.g. pH – hydrogen potential

Line 568          Please harmonize within the manuscript Abbreviation of journal name or not!

Line 606          Please harmonize within the manuscript Abbreviation of journal name or not!

Line 611          Please harmonize within the manuscript Abbreviation of journal name or not!

Line 660          Please harmonize within the manuscript Abbreviation of journal name or not!

Author Response

Dear Reviewer,

Thank you very much for your comments and suggestions. All changes in the manuscript are highlighted in yellow and tracked changes.

Best regards,

Prof. dr. Ramunas Antanaitis

 Comments and answers

Comments: Line 8              Some of the hyperlinks are not correct (e.g. „.lt“ is missing). Please correct

Answers: Corrected

Comments: Line 21            The term „acid base-balance“ is written in a different font size.

Answers: Corrected

Comments: Line 33            Please add a full stop between 2023.06.15 and „-„ and between „-„ and 06.23)

Answers: Corrected

Comments: Line 52            Please delete the blanket space after „acid-base balance“

Answers: Deleted

Comments: Line 52            Please delete the blanket space before „predicted behaviors“

Answers: Deleted

Comments: Line 61            Please delete one blanket space between „.“ and „At the point“

Answers: Deleted

Comments: Line 87            There is missing a full stop at the end of the sentence

Answers: Corrected

Comments: Line 107          Please add a blanket space between „assessments“ and [13].

Answers: Corrected

Comments: Line 107          What does „OBJ“ mean?

Answers: Deleted

Comments: Line 112          Please harmonize within the manuscript: References with year numbers or not?

Answers: Deleted year numbers

Comments: Line 140          Please delete one blanket space between „cattle.“ and „To possibly“

Answers: Deleted

Comments: Line 158          Please add the coordinates.

Answers: We added information – “coordinates - 55.819156, 23.773541)”

Comments: Line 178          Please harmonize within the manuscript: 0600 or 06:00?

Answers: Corrected to – “06:00 and 18:00 h”

Comments: Line 179          Please delete the full stop after 36 kg/d.

Answers: Deleted

Comments: Line 180          Please harmonize within the manuscript blanket space between value and unit (%) or not?

Answers: Corrected to – “ f.e. - 3.6 %”

Comments: Line 182          Please delete one blanket space between „farm.“ and „The current study“

Answers: Corrected

Comments: Line 220          Please specify: Which 50 cows?

Answers: We corrected to – “Blood was drawn from each of the 50 cows, which were selected from 350 Holstein cows (in their second or subsequent lactation period), using the jugular venipuncture technique.”

Comments: Line 239          The abbreviation LSD was not introduced before

Answers: We corrected to – “The least significant difference (LSD)..”

Comments: Line 241          Please harmonize within the manuscript: P<0.05 or P < 0.05?

Answers: Corrected to – “P<0.05”

Comments: Line 247          Please add a blanket space between THI and >78

Answers: We corrected to – “(THI > 78).

Comments: Line 249          Please add a blanket space between < and 72

Answers: We corrected to – “(THI < 72).”

Comments: Table 3           Tables should be self-explanatory, so the table heading should also include relevant information about the statistical analysis.

Answers: We corrected to – “Table 3. The changes in data regarding rumination, drinking, and locomotory behavior across different risk scenarios of heat stress”

Comments: Line 259          Please add a blanket space between THI and >78

Answers: We corrected to – “(THI > 78)”

Comments: Line 259          Please add a blanket space between < and 72

Answers: We corrected to – “(THI < 72)”

Comments: Figure 1          Please use the same font as in the text

Answers: Corrected

     Comments:  Sometimes the legend is not correct (THI 1) instead of (TH I1)

                        F is not formatted like the others (hours)

                        Mean of chews per minute

Answers: Corrected

Comments: Table 4           footnotes should be formatted similar: e.g. Rumination time – RT

Answers: Corrected to – “Correlation is significant. * P<0.05. OAT - other activity time; RT- rumination time; EDT - eat down time; EUT- eat up time; DT - drink time; OC - other chew; CR chew rumination; EDC - eat down chews; EUC - eat up chews; G - gulp; B - bolus; CPM - chews per minute; CPB - chews per bolus; WT - walking time; UT - up time; DT - down time; AC - activity change.”

Comments: Table 5           THI >78 instead of THI>78

Answers: Corrected to – “THI >78

 Comments:  footnotes should be formatted similar: e.g. pH – hydrogen potential

Answers: corrected

Comments: Line 345          Please add a blanket space between „-„ and „urea“

Answers: Corrected to – “– urea”

Comments: Line 346          Please add a blanket space between „THI“ and „3“

Answers: Corrected to – “THI 3”

Comments: Figure 2          footnotes should be formatted similar: e.g. pH – hydrogen potential

Answers: Corrected

Comments: Line 568          Please harmonize within the manuscript Abbreviation of journal name or not!

Answers: Corrected to – “Compendium on Continuing Education for the Practicing Veterinarian

Comments: Line 606          Please harmonize within the manuscript Abbreviation of journal name or not!

Answers: Corrected to – “Scientific Reports

Comments: Line 611          Please harmonize within the manuscript Abbreviation of journal name or not!

Answers: Corrected to – “Asian-Australasian Journal of Animal Sciences

Comments: Line 660          Please harmonize within the manuscript Abbreviation of journal name or not!

Answers: Corrected to – “International Journal of Biometeorology”

Reviewer 2 Report (Previous Reviewer 3)

Comments and Suggestions for Authors

The experiment is interesting but it seems to me that author should keep the manuscript in stand-by for a while because it needs a fresh restart. There is some issue with the experimental design and dates that doesn't match and  makes the experiment unclear. Similarly, I have some concerns with the statistics.

In my opinion, the sentence "We used the THI to divide the cows under investigation into 3 groups..." is confousing. You divided the experimental period in three sub-periods, but the cows are the same in the three groups. Maybe I am missing something, but it is pretty important clarify the basis of the experiment.

According to the M&M, the collection of data of High HS last 3 days, not a week, because the adaptation period overlaps with this group.

L189-192 and L247-249 repeated

L158 and L182 repeated numbar ov cows

L150 coordinates are missing

Figure 1. Needs a clear legend. What are THI1, THI2 and THI3?

Table 5 and Figure 2. You studied the influence of the THI on the variables. What is the sense of calculate correlations with the levels of THI? If you want to study the effect of THI may be you should try linear contrast, no correlations...

Author Response

Dear Reviewer,

Thank you very much for your comments and suggestions. All changes in the manuscript are highlighted in yellow and tracked changes.

Best regards,

Prof. dr. Ramunas Antanaitis

Comments: The experiment is interesting but it seems to me that author should keep the manuscript in stand-by for a while because it needs a fresh restart. There is some issue with the experimental design and dates that doesn't match and  makes the experiment unclear. Similarly, I have some concerns with the statistics.

Comments:In my opinion, the sentence "We used the THI to divide the cows under investigation into 3 groups..." is confousing. You divided the experimental period in three sub-periods, but the cows are the same in the three groups. Maybe I am missing something, but it is pretty important clarify the basis of the experiment.

Answers: We corrected to – “We used the THI to divide the cows under investigation into 3 periods [27]: I. high HS—THI > 78 (period: 2023.06.15- 06.23); II. medium HS—THI 72-78 (period: 2023.06.24 – 06.30); and III. low HS—THI < 72 (period: 2023.07.01- 08)”

Comments: According to the M&M, the collection of data of High HS last 3 days, not a week, because the adaptation period overlaps with this group.

Answers: We corrected to – “We used the THI to divide the cows under investigation into 3 periods [27]: I. high HS—THI > 78 (period: 2023.06.15- 06.23); II. medium HS—THI 72-78 (period: 2023.06.24 – 06.30); and III. low HS—THI < 72 (period: 2023.07.01- 08).

The RWS tests were performed between 2023.06.15 and 2023.07.30. The cows under investigation acclimatized to the RWS during the adaptation period, which was set from 01 to 15 June 2023. The registration process started on 2023.06.15 and terminated on 2023.07.08, and was performed every hour, 24 hours per day.”

Comments: L189-192 and L247-249 repeated

Answers: We deleted second sentence.

Comments: L158 and L182 repeated numbar ov cows

Answers: We deleted second sentence.

Comments: L150 coordinates are missing

Answers: We added – “coordinates - 55.819156, 23.773541)”

Comments: Figure 1. Needs a clear legend. What are THI1, THI2 and THI3?

Answers:

The remark below figure was added: 3 periods of temperature humidity index: THI 1, THI 2, THI 3.

Comments: Table 5 and Figure 2. You studied the influence of the THI on the variables. What is the sense of calculate correlations with the levels of THI? If you want to study the effect of THI may be you should try linear contrast, no correlations...

Answers:

The calculation of correlation shows the strength of the relationship, also the direction is it positive or is it negative.

We don’t think that linear contrast calculation will better represent the data.

ANOVA

Sum of Squares

df

Mean Square

F

Sig.

pH

Between Groups

(Combined)

,018

2

,009

3,233

,055

Linear Term

Contrast

,016

1

,016

5,847

,023

Deviation

,002

1

,002

,620

,438

Within Groups

,073

27

,003

Total

,091

29

pCO2

Between Groups

(Combined)

233,867

2

116,933

3,279

,053

Linear Term

Contrast

192,200

1

192,200

5,389

,028

Deviation

41,667

1

41,667

1,168

,289

Within Groups

962,923

27

35,664

Total

1196,790

29

pO2

Between Groups

(Combined)

13047,923

2

6523,961

3,485

,045

Linear Term

Contrast

542,882

1

542,882

,290

,595

Deviation

12505,041

1

12505,041

6,680

,015

Within Groups

50544,007

27

1872,000

Total

63591,930

29

cHCO3-

Between Groups

(Combined)

3,701

2

1,850

,526

,597

Linear Term

Contrast

,480

1

,480

,136

,715

Deviation

3,220

1

3,220

,915

,347

Within Groups

95,046

27

3,520

Total

98,747

29

BE (ecf)

Between Groups

(Combined)

3,272

2

1,636

,429

,655

Linear Term

Contrast

1,922

1

1,922

,504

,484

Deviation

1,350

1

1,350

,354

,557

Within Groups

102,880

27

3,810

Total

106,152

29

cSO2

Between Groups

(Combined)

18,721

2

9,360

3,838

,034

Linear Term

Contrast

,128

1

,128

,052

,821

Deviation

18,593

1

18,593

7,623

,010

Within Groups

65,854

27

2,439

Total

84,575

29

Na+

Between Groups

(Combined)

7,200

2

3,600

1,062

,360

Linear Term

Contrast

7,200

1

7,200

2,125

,156

Deviation

,000

1

,000

,000

1,000

Within Groups

91,500

27

3,389

Total

98,700

29

K+

Between Groups

(Combined)

,133

2

,066

,644

,533

Linear Term

Contrast

,002

1

,002

,019

,890

Deviation

,131

1

,131

1,268

,270

Within Groups

2,782

27

,103

Total

2,915

29

Ca++

Between Groups

(Combined)

,001

2

,000

,104

,901

Linear Term

Contrast

,000

1

,000

,078

,783

Deviation

,001

1

,001

,131

,720

Within Groups

,111

27

,004

Total

,112

29

Cl-

Between Groups

(Combined)

5,267

2

2,633

,693

,509

Linear Term

Contrast

5,000

1

5,000

1,316

,261

Deviation

,267

1

,267

,070

,793

Within Groups

102,600

27

3,800

Total

107,867

29

TCO2

Between Groups

(Combined)

4,611

2

2,305

,650

,530

Linear Term

Contrast

1,058

1

1,058

,298

,589

Deviation

3,553

1

3,553

1,002

,326

Within Groups

95,699

27

3,544

Total

100,310

29

Hct

Between Groups

(Combined)

6,067

2

3,033

1,147

,333

Linear Term

Contrast

6,050

1

6,050

2,288

,142

Deviation

,017

1

,017

,006

,937

Within Groups

71,400

27

2,644

Total

77,467

29

cHgb

Between Groups

(Combined)

,515

2

,257

,968

,393

Linear Term

Contrast

,512

1

,512

1,925

,177

Deviation

,003

1

,003

,010

,921

Within Groups

7,180

27

,266

Total

7,695

29

BE(b)

Between Groups

(Combined)

3,413

2

1,706

,542

,588

Linear Term

Contrast

2,664

1

2,664

,847

,366

Deviation

,748

1

,748

,238

,630

Within Groups

84,966

27

3,147

Total

88,379

29

Glu

Between Groups

(Combined)

1,049

2

,524

5,148

,013

Linear Term

Contrast

,968

1

,968

9,504

,005

Deviation

,081

1

,081

,792

,381

Within Groups

2,750

27

,102

Total

3,799

29

Lac

Between Groups

(Combined)

3,190

2

1,595

10,161

<,001

Linear Term

Contrast

2,911

1

2,911

18,546

<,001

Deviation

,279

1

,279

1,776

,194

Within Groups

4,238

27

,157

Total

7,427

29

BUN

Between Groups

(Combined)

37,267

2

18,633

3,491

,045

Linear Term

Contrast

24,200

1

24,200

4,534

,042

Deviation

13,067

1

13,067

2,448

,129

Within Groups

144,100

27

5,337

Total

181,367

29

Urea

Between Groups

(Combined)

4,958

2

2,479

3,763

,036

Linear Term

Contrast

3,120

1

3,120

4,736

,038

Deviation

1,837

1

1,837

2,789

,106

Within Groups

17,789

27

,659

Total

22,747

29

Crea

Between Groups

(Combined)

675,267

2

337,633

4,249

,025

Linear Term

Contrast

661,250

1

661,250

8,321

,008

Deviation

14,017

1

14,017

,176

,678

Within Groups

2145,700

27

79,470

Total

2820,967

29

BUN/Crea

Between Groups

(Combined)

19,741

2

9,870

1,021

,374

Linear Term

Contrast

,018

1

,018

,002

,966

Deviation

19,723

1

19,723

2,040

,165

Within Groups

260,981

27

9,666

Total

280,722

29

According to 3 periods of temperature humidity index (THI1, THI2, THI3)

Contrast Tests

Contrast

Value of Contrast

Std. Error

t

df

Sig. (2-tailed)

95% Confidence Interval

Lower

Upper

pH

Assumes equal variances

1

44,72040a

,061710

724,689

27

<,001

44,59378

44,84702

Does not assume equal variances

1

44,72040a

,057223

781,517

21,089

<,001

44,60143

44,83937

pCO2

Assumes equal variances

1

250,820a

7,0661

35,496

27

<,001

236,322

265,318

Does not assume equal variances

1

250,820a

6,9454

36,113

19,304

<,001

236,299

265,341

pO2

Assumes equal variances

1

1106,640a

51,1938

21,617

27

<,001

1001,599

1211,681

Does not assume equal variances

1

1106,640a

46,7122

23,691

14,077

<,001

1006,504

1206,776

cHCO3-

Assumes equal variances

1

174,490a

2,2200

78,600

27

<,001

169,935

179,045

Does not assume equal variances

1

174,490a

2,3698

73,630

15,241

<,001

169,446

179,534

BE (ecf)

Assumes equal variances

1

30,860a

2,3097

13,361

27

<,001

26,121

35,599

Does not assume equal variances

1

30,860a

2,2436

13,755

19,018

<,001

26,164

35,556

cSO2

Assumes equal variances

1

594,840a

1,8479

321,904

27

<,001

591,048

598,632

Does not assume equal variances

1

594,840a

1,7081

348,254

9,111

<,001

590,983

598,697

Na+

Assumes equal variances

1

816,60a

2,178

374,901

27

<,001

812,13

821,07

Does not assume equal variances

1

816,60a

2,109

387,251

20,264

<,001

812,20

821,00

K+

Assumes equal variances

1

24,660a

,3798

64,928

27

<,001

23,881

25,439

Does not assume equal variances

1

24,660a

,3539

69,687

18,712

<,001

23,919

25,401

Ca++

Assumes equal variances

1

7,1060a

,07599

93,510

27

<,001

6,9501

7,2619

Does not assume equal variances

1

7,1060a

,06807

104,395

20,377

<,001

6,9642

7,2478

Cl-

Assumes equal variances

1

608,60a

2,307

263,862

27

<,001

603,87

613,33

Does not assume equal variances

1

608,60a

2,267

268,442

19,958

<,001

603,87

613,33

TCO2

Assumes equal variances

1

173,160a

2,2276

77,734

27

<,001

168,589

177,731

Does not assume equal variances

1

173,160a

2,3927

72,371

15,058

<,001

168,062

178,258

Hct

Assumes equal variances

1

156,30a

1,924

81,232

27

<,001

152,35

160,25

Does not assume equal variances

1

156,30a

1,884

82,981

19,837

<,001

152,37

160,23

cHgb

Assumes equal variances

1

53,200a

,6102

87,190

27

<,001

51,948

54,452

Does not assume equal variances

1

53,200a

,6183

86,041

19,359

<,001

51,907

54,493

BE(b)

Assumes equal variances

1

28,170a

2,0990

13,421

27

<,001

23,863

32,477

Does not assume equal variances

1

28,170a

1,9991

14,092

19,973

<,001

24,000

32,340

Glu

Assumes equal variances

1

19,600a

,3776

51,905

27

<,001

18,825

20,375

Does not assume equal variances

1

19,600a

,2720

72,062

26,704

<,001

19,042

20,158

Lac

Assumes equal variances

1

8,7250a

,46876

18,613

27

<,001

7,7632

9,6868

Does not assume equal variances

1

8,7250a

,39738

21,956

23,216

<,001

7,9034

9,5466

BUN

Assumes equal variances

1

77,20a

2,733

28,243

27

<,001

71,59

82,81

Does not assume equal variances

1

77,20a

2,581

29,909

19,390

<,001

71,80

82,60

Urea

Assumes equal variances

1

27,470a

,9604

28,602

27

<,001

25,499

29,441

Does not assume equal variances

1

27,470a

,8987

30,566

19,846

<,001

25,594

29,346

Crea

Assumes equal variances

1

448,30a

10,548

42,501

27

<,001

426,66

469,94

Does not assume equal variances

1

448,30a

12,432

36,060

13,834

<,001

421,61

474,99

BUN/Crea

Assumes equal variances

1

92,560a

3,6786

25,162

27

<,001

85,012

100,108

Does not assume equal variances

1

92,560a

3,6791

25,158

19,429

<,001

84,871

100,249

a. The sum of the contrast coefficients is not zero.

Reviewer 3 Report (New Reviewer)

Comments and Suggestions for Authors

Thank you for addressing the suggested modifications. 

Author Response

Comments and Suggestions for Authors

Reviewer: Thank you for addressing the suggested modifications. 

Authors: Dear Reviewer, We are thankful for your review.

This manuscript is a resubmission of an earlier submission. The following is a list of the peer review reports and author responses from that submission.

Round 1

Reviewer 1 Report

Comments and Suggestions for Authors

The present paper studies the effect of the heat stress on the blood gas parameters and on the behavior (rumination, drinking and locomotory) of 350 dairy cows. Because of the attention given in the last years to the role of sensors and other technologies applied to the monitoring of cows’ welfare in dairy farms, the analyzed topic could be of great interest. However, the paper fails to 

INTRODUCTION: too long and too heavy to read. It should be reduced, also considering that the more specific info is used or should be used in the discussion of results.

Some sentences lack references (L58, L72, L79, L86, L89-97, L110)

MATERIALS AND METHODS: this section is well explained, but should be improved by the addition of some info. For example how the 3 thresholds of THI have been established (<72, 72-78, >78)? Are there other studies using these specific ranges? 

Why did you collect blood samples only during the acclimation phase?

What do you mean by horizontal and vertical angles? L210

The statistical analysis should be improved in its description and use. For example, it would be interesting to know how many RWS data per animal per hour have been used for the ANOVA analysis, in order to understand the data editing done before. Moreover, simply reporting the ranges of the RWS parameters over 24h without comparing them to daylight hours and/or ambient temperature/THI per hour is a waste. It gives you the variability of the trait avoiding to explain this variability with the changes of the surrounding environment.

The factors used for the ANOVA and linear regression are not clearly explained. The individual variability was not considered as a factor during the analysis but it was cited at line 440. It is not clear if the heat stress corresponds to the heat index and how it is calculated (line 184). Moreover, the THI corresponds to the three weeks of the monitoring so the differences between the 3 levels of THI factors can be due to other variables not considered in this study.

RESULTS The authors insist that both RWS and blood parameters have the highest statistically significant mean differences between THI weeks, but only few RWS (7 over 17) and blood (4 over 20) parameters resulted significantly affected by the THI change.    

DISCUSSION: redundant with introduction

CONCLUSION: barely reflects what really exposed in the results.

Other minor comments are listed below.

L12: authors email in the wrong section

L16: HS should be defined above

L19: check for bold font

L31: HS should be defined above

L34-35:”…

L87 what do you mean by “summer stress”?

L137: Gianesella, please check authors name spelling

L127: please use the acronym PDF

L143 SARA and PO2 are introduced without the full names

L187-188: provide bibliography for the THI range choice

L186 after “3” is missing “groups”

L221-22:redundant sentence

L224 wrong chemical annotation of compounds

L234: change “during” in “with”

Table 1: in significant results, if the levels go from “a” to “c”, then what does the “d” letter refer to?

L255-259: the results “ranged” more than “changed”

L277-278: I think you confused the results. It is the week with THI>78 that had the highest urea level, not the THI<72 week.

L270-279: it is difficult to read, please rephrase.

Figure 1 missed the label of y-axis

Comments on the Quality of English Language

The manuscript needs an extensive English revision to improve the clearness 

Author Response

Dear Reviewer,  

Authors are very thankful for the comments, which help us to improve the manuscript. All changes proposed have been included in the manuscript and highlighted in yellow and track changes.   

Best Regards,  

Prof. Ramunas Antanaitis 

You can find your questions and comments along with our responses here: 

INTRODUCTION: too long and too heavy to read. It should be reduced, also considering that the more specific info is used or should be used in the discussion of results. 

Answer: we corrected the whole introduction section.  

Some sentences lack references (L58, L72, L79, L86, L89-97, L110) 

Answer: we corrected the whole introduction section. 

MATERIALS AND METHODS: this section is well explained, but should be improved by the addition of some info. For example how the 3 thresholds of THI have been established (<72, 72-78, >78)? Are there other studies using these specific ranges?  

Answer: we added information - “We used the THI to divide the cows under investigation into 3 groups [21]:...” 

Gantner, V.; Mijić, P.; Kuterovac, K.; Solić, D.; Gantner, R. Temperature-Humidity Index Values and Their Significance on the Daily Production of Dairy Cattle. Mljekarstvo : časopis za unaprjeđenje proizvodnje i prerade mlijeka 2011, 61, 56–63. 

Why did you collect blood samples only during the acclimation phase? 

Answer: sorry, it was a mistake, we corrected - “The blood gas parameters were registered from 20 to 30 June 2023, once per week”   

What do you mean by horizontal and vertical angles? L210 

Answer: we corrected - “up time (min/h)—when the pedometer changes its position from a horizontal to vertical angle for a duration of at least 50 s; down time (min/h)—when the pedometer angle changes its position from a vertical to horizontal angle for a duration of at least 50 s”  

The statistical analysis should be improved in its description and use. For example, it would be interesting to know how many RWS data per animal per hour have been used for the ANOVA analysis, in order to understand the data editing done before. Moreover, simply reporting the ranges of the RWS parameters over 24h without comparing them to daylight hours and/or ambient temperature/THI per hour is a waste. It gives you the variability of the trait avoiding to explain this variability with the changes of the surrounding environment.  

Answer:  records for RWS data per animal per hour have been used (it was mentioned in the section Assessments of rumination, drinking, and locomotory parameters using RumiWatch sensors). 

We have made corrections (Figure 1); the new figures are added according to THI and the change of investigated traits per day (24 h). 

The factors used for the ANOVA and linear regression are not clearly explained. The individual variability was not considered as a factor during the analysis but it was cited at line 440. 

Answer: THI week and cow Id was added in Repeated measures Anova analysis – but the representation of data in our opinion is not very informative, because it shows a lot of means ( for 10 cows (repeated measures of 10 cows) and for each trait.  The results of interaction THI*Cow Id  in a Pairwise comparison of means we receive very large and cluttered uninformative table, the same large table for the means, each cow during 3 THI weeks for each trait. We don’t think that it is a good way of data representation. Also we believe in order to make General linear model for such interactions the sample size is too low. 

Please explain maybe we don’t understand, how we should express the data with variability of cows. 

 It is not clear if the heat stress corresponds to the heat index and how it is calculated (line 184). Moreover, the THI corresponds to the three weeks of the monitoring so the differences between the 3 levels of THI factors can be due to other variables not considered in this study. 

Answer: The temperature humidity index (THI) was measured on the farm using a SmaXtec climate sensor (SmaXtec animal care GmbH, Graz, Austria). The formula used to calculate the temperature humidity index was THI = 0.8 × T + RH × (T 14.4) + 46.4 [21]. The heat index was calculated using a heat stress calculator (SmaXtec animal care GmbH, Graz, Austria). We used the THI to divide the cows under investigation into 3 groups [21]: I. high risk of HS—THI > 78 (period: 2023.06.15- 06.23); II. medium risk of HS—THI 72-78 (period: 2023.06.24 – 06.30); and III. low risk pf HS—THI < 72 (period: 2023.07.01- 08). 

RESULTS The authors insist that both RWS and blood parameters have the highest statistically significant mean differences between THI weeks, but only few RWS (7 over 17) and blood (4 over 20) parameters resulted significantly affected by the THI change.     

Answer: we corrected to - “The data analysis of the RWS parameters presented the highest statistically significant mean differences between the other activity time, other chewing, activity change parameters, during different risk scenarios of HS (Table 1)”  

DISCUSSION: redundant with introduction 

Answer: We have rewritten the entire discussion section  

CONCLUSION: barely reflects what really exposed in the results. 

Answer: We have rewritten the entire conclusion section -  

“We can conclude that heat stress significantly impacts activity time, chewing behaviors, and general activity, as registered by innovative technologies, as well as blood gas parameters in fresh multiparous dairy cows. We found that specifically during the high heat stress (HS) risk period (THI > 78), there was an 11.75% increase in other activity time and a 17.67% increase in other chewing behaviors compared to the low-risk period (THI < 72). Even in the medium HS risk period (THI 72–78), these behaviors were elevated (13.80% increase in chewing behaviors) compared to low-risk conditions. Interestingly, a 12.82% increase in activity change was observed when transitioning from medium to low HS risk. We observed a substantial and statistically significant increase in BUN levels during weeks with higher THI values (>78) compared to those with moderate THI levels (72–78). Specifically, BUN levels were 19.75% higher in weeks where THI exceeded 78, compared to weeks with THI values between 72 and 78. This increase is noteworthy as it was 16.89% higher than the urea level for the same period. Moreover, when comparing the highest THI values (>78) to the second and third weeks, urea levels were found to be 21.16% and 18.81% higher, respectively.  

Based on our study's findings, we advise dairy farmers to employ innovative technologies for the real-time monitoring and management of heat stress in cows. During periods of high THI (> 78), it is crucial to closely monitor changes in activity time and chewing behaviors, as significant increases in these behaviors are indicative of elevated heat stress levels. Farmers should also adjust their nutritional plans based on continuous monitoring of blood urea nitrogen levels. This innovative approach, leveraging advanced technologies, is vital for maintaining the health and productivity of dairy cows under various climatic conditions”  

Other minor comments are listed below. 

L12: authors email in the wrong section 

Answer: we corrected - [email protected]   

L16: HS should be defined above 

Answer: we corrected to - “Heat stress (HS) is a factor that..”  

L19: check for bold font 

Answer: corrected  

L31: HS should be defined above 

Answer: corrected  

L87 what do you mean by “summer stress”? 

Answer: we changed “summer stress” to “HS”  

L137: Gianesella, please check authors name spelling 

Answer: Corrected to - “Gianesella..”  

L127: please use the acronym PDF 

Answer: we corrected to - “Applying technologies, like RumiWatch, can provide dairy breeders with important information on how to reduce the negative effects of HS on cow health”  

L143 SARA and PO2 are introduced without the full names 

Answer: Corrected  

L187-188: provide bibliography for the THI range choice 

Answer: we added - “We used the THI to divide the cows under investigation into 3 groups [21]:..” 

Gantner, V.; Mijić, P.; Kuterovac, K.; Solić, D.; Gantner, R. Temperature-Humidity Index Values and Their Significance on the Daily Production of Dairy Cattle. Mljekarstvo : časopis za unaprjeđenje proizvodnje i prerade mlijeka 2011, 61, 56–63. 

L186 after “3” is missing “groups” 

Answer: we corrected to - “We used the THI to divide the cows under investigation into 3 groups [21]:..” 

L221-22:redundant sentence 

Answer: we deleted this sentence.  

L224 wrong chemical annotation of compounds 

Answer: corrected  

L234: change “during” in “with” 

Answer: corrected to - “..with THI weeks..” 

Table 1: in significant results, if the levels go from “a” to “c”, then what does the “d” letter refer to? 

Answer: Corrected – we deleted “d” 

L255-259: the results “ranged” more than “changed” 

Answer: we corrected to - “The mean values for the eating time changes assessed every 24 h showed that they ranged from 4.21 to 13.61 min/h; the drinking time ranged from 0.20 to 0.52 min/h; chews per bolus ranged from 2.91 to 7.94 n/min; and up time ranged from 4.24 to 17.03 min/h; other activity ranged from 16.56 to 27.19; other chew ranged from 87.77 to 173.72 min/h; chews per minute ranged from 50.77 to 73.41 n/min; activity ranged from 60.99 to 121.82 min/h; rumination time ranged from 15.19 to 29.9 min/h; bolus ranged from 16.4 to 31.64 n/h; and down time ranged from 16.48 to 24.95 min/h (Figure 1A, B and C)”  

L277-278: I think you confused the results. It is the week with THI>78 that had the highest urea level, not the THI<72 week. 

Answer: we corrected to - “The urea level was 21.16 percent higher in the week presenting a THI value of THI>78..”  

L270-279: it is difficult to read, please rephrase. 

Answer:  

Figure 1 missed the label of y-axis 

Answer: Corrected 

Reviewer 2 Report

Comments and Suggestions for Authors

General comments:

The manuscript animals-2784451, entitled "The Impacts of Heat Stress on the Factors of Rumination, Drinking, and Locomotory Behavior in Dairy Cows, Registered by Innovative Technologies and Blood Gas Parameters" with Mr. Antanaitis as first author deals with an interesting topic within the emerging field of animal welfare in dairy cattle. The manuscript is relatively structured, in most cases carefully formulated and presents interesting ideas. However, some sections need to be revised and major mistakes must be corrected especially in the results and discussion section. Various comments for changes to the text could be find in the following section.

Detailed comments:

 Line 4              In my view, the methods used are interesting, but not very innovative. I would therefore use a somewhat more neutral term in the title

Line 8              Please delete “1” in the e-mail address.

Line 12            Please insert the complete links for all e-mail addresses.

Line 12            The e-mail adress of Giedrius Palubinskas is not correct.

Line 12            Please insert a comma between “(L.A.)” and “giedrius…”

Line 16            The abbreviation “HS” is not introduced. Please use the full term or introduce it before.

Line 21            I would suggest to change the order in this sentence: blood gas parameters, rumination, eating and drinking, locomotory patterns

Line 21            Which kind of decreasing rumination? Time, frequency? Please add more details.

Line 28            Please rephrase this sentence: …in a Lithuanian dairy farm with 850 milking cows.

Line 33            The date should be corrected: (period: 2023.07.01. – 2023.07.08.)

Line 33            Please harmonize within the manuscript: “Rumiwatch” or “ RumiWatch”?

Line 38            How can the parameters be measured once a week if the entire period only lasted 5 days? Wouldn't it have made more sense to examine several blood samples in the different collection periods?

Line 40            Are the differences between the weeks expressed as percentages or percentage points?

Line 44            What activities are meant here? Please provide more details regarding the definition.

Line 45            This sentence is too long and, in my view, contains some duplications. Please shorten the sentence and focus on the most important facts.

Line 51            The abstract does not contain any results on blood gas values. So why do you refer to it?

Line 53            The term global warming is too general. Use another term that describes “heat stress”. As already mentioned the methods used are not very innovative. Please use another term. Please add the terms “blood gas parameters” and “behavior parameters” within the key words.

Line 55            General comment: The introduction should be rewritten, as it is very unstructured and does not lead the reader to the topic as intended. I would suggest starting with a short section on climate change and the expected consequences for the dairy industry. Then it would be useful to describe some of the effects of heat stress on the behavior and health of cows (e.g. feed intake, water intake, rumination, activities). Finally, the various methods that can be used to identify these changes (e.g. sensors, blood gas parameters) could be presented.

Line 57            The results of this study should not be cited in the introduction as justification for the usefulness of the study. The research gap should be identified on the basis of a comprehensive literature review.

Line 77            The sentence is too long. Please split into two sentences.

 Line 78            Please insert the term “they are” between “when” and “outside”.

 Line 83            The abbreviation “DMI” was not introduced before. Please write out the full term the first time it is mentioned

 Line 105          Please use the term “Precision livestock farming”, because the following explanations are not limited to cows.

Line 123          Are there really so few studies on the subject? Research has been going on for many years on the subject of "heat stress" and some groups will certainly have used this type of sensor. Please cite some original papers to specifically identify the research gap.

Line 131          Comments on the need for further research actually belong in the conclusions and not in the introduction. Please move this section to the other chapter.

Line 142          The abbreviations in the following sentences were not introduced before. Please write out the full term the first time it is mentioned.

Line 157          The objective of the study remains unclear. Please specify the objectives of the study and emphasize the novelty value.

Line 168          Why was the ration calculated for a 500 kilogram cow when the herd weighs 600 kilograms on average?

Line 180          Why were only 350 cows examined and primiparous cows not included?

Line 183          Please justify the selection of this THI formula and the associated threshold values. There are different views on the use of the respective formulas in the scientific debate.

Line 186          Please insert the term “groups” between “3” and “[20]”.

Line 186          23.06.2023 is included in the first two survey periods. Why have you not made a clear separation of the data sets here?

Line 199          Please give an example of “other jaw movements”.

Line 214          Maybe “rumination, feeding or drinking activity”?

Line 235          Please harmonize within the manuscript “p” or “P”; and “p<0.05” or “p < 0.05”!

Line 238          There is no reference to the presentation of the data in Table 1.

Line 245          Please delete the second bracket after “P<0.001”.

Table 1           Please widen the table to make the contents easier to read. Tables should be self-explanatory, so abbreviations should be written out in full or integrated into footnotes. Use the same number of decimal places in the table. Why do you use the superscript d in addition to a,b,c? What is the significance of this? Why is there an asterisk in the line "other chew"? I would sort the rows according to the individual behaviors (rumination, eating, activity, etc.). This makes it easier to compare the results with each other. Why is the p-value written in italics?

Line 255          There is no reference to the presentation of the data in Figure 1. Please add the units of the values.

Figure 1          The units on the y-axis are missing. Changes in the behavior of cows during the course of the day have been known for a long time and are mainly influenced by milking times and feeding. Wouldn't it make more sense here to limit ourselves to a few parameters and compare these between the survey periods? For example, how does feeding behavior change during heat stress compared to cooler periods? Do the animals eat more during the cooler night and should the feeding times be adjusted accordingly?

Line 271          The abbreviation “BUN” was not introduced before. Please write out the full term the first time it is mentioned

Line 275          Please replace the comma through a full stop (16.89).

Table 2           Tables should be self-explanatory, so abbreviations should be written out in full or integrated into footnotes. Why do you use the superscript d in addition to a,b,c? What is the significance of this? Why is the p-value written in italics?

Line 284          Please insert “gas parameters” after the term “RWS parameters and blood”

Line 285          This is already part of the discussion and should not be mentioned in the results chapter.

Table 3           Please widen the table to make the contents easier to read. Tables should be self-explanatory, so abbreviations should be written out in full or integrated into footnotes.

Line 297          Why is it an attempt of the industry? I think it is a natural bevioral change of HS cows?

Line 205          Please harmonize within the manuscript “25 °C” or “25°C”.

Line 308          The abbreviation HS was introduced before and must not be mentioned again

Line 311          Please cite some studies using also THI thresholds instead of ° Celsius in order to compare the results. Alternatively you can add some more details concerning temperature and humidity in the Material and Method chapter.

Line 316          The abbreviation “TNZ” was not introduced before. Please write out the full term the first time it is mentioned

Line 321          Why did you use RWS as an abbreviation for real-time welfare scoring?

Line 325          However, in your study there is no linear relationship between heat stress and walking time!

Line 344          Please insert a paragraph here.

Line 344          This section does not promote knowledge gain. I would again suggest to focus on behavioral alterations between the three groups.

Line 371          Please insert a reference for this statement.

Line 373          Please discuss in more detail why there are no connections between the RWS indicators and the blood gas parameters. Is it because of the methodology or do you generally not suspect a connection? What does this lack of connection mean for the usefulness of the RWS indicators?

Line 405          Please go into detail about the practical benefits of the work. How can farmers benefit from your insights?

Line 415          Please do not use the term “robust statistical analysis” if you rightly pointed out the small sample size beforehand

Line 445          The name of the first author is not correct.

Line 497          Maybe “Animal Science Papers” instead of “Animal science papers”?

Line 521          Please delete “1” after the title.

Line 525          The names of the authors are not correct.

Line 533          Maybe “Animal” instead of “animal”?

Line 535          Please delete “2” after the title.

Line 541          Please delete “2,3” after the title.

Author Response

Dear Reviewer,  

Authors are very thankful for the comments, which help us to improve the manuscript. All changes proposed have been included in the manuscript and highlighted in yellow and track changes.   

Best Regards,  

Prof. Ramunas Antanaitis 

You can find your questions and comments along with our responses here: 

General comments: 

The manuscript animals-2784451, entitled "The Impacts of Heat Stress on the Factors of Rumination, Drinking, and Locomotory Behavior in Dairy Cows, Registered by Innovative Technologies and Blood Gas Parameters" with Mr. Antanaitis as first author deals with an interesting topic within the emerging field of animal welfare in dairy cattle. The manuscript is relatively structured, in most cases carefully formulated and presents interesting ideas. However, some sections need to be revised and major mistakes must be corrected especially in the results and discussion section. Various comments for changes to the text could be find in the following section. 

Answer: Thank you very much for your valuable and constructive feedback on our manuscript. Your insights and suggestions have been instrumental in enhancing the quality and clarity of our work. We greatly appreciate the time and effort you have dedicated to reviewing our study, and we look forward to incorporating your suggestions in our revised manuscript. 

Detailed comments: 

Line 4 In my view, the methods used are interesting, but not very innovative. I would therefore use a somewhat more neutral term in the title 

Answer:  

Line 8 Please delete “1” in the e-mail address. 

Answer: We corrected to - [email protected]  

Line 12 Please insert the complete links for all e-mail addresses. 

Answer: We corrected to - [email protected]  

Line 12 The e-mail adress of Giedrius Palubinskas is not correct. 

Answer: We corrected to - [email protected] 

Line 12 Please insert a comma between “(L.A.)” and “giedrius…” 

Answer: Corrected  

Line 16 The abbreviation “HS” is not introduced. Please use the full term or introduce it before. 

Answer: we corrected to - “Heat stress (HS)..” 

Line 21 I would suggest to change the order in this sentence: blood gas parameters, rumination, eating and drinking, locomotory patterns 

Answer: we corrected this sentence to - “This study suggests that heat stress has a negative impact on dairy cows, resulting in considerable decreases in their eating and drinking habits, reduction in rumination time, and alterations in their locomotion patterns as well as in their blood gas parameters.”  

Line 21 Which kind of decreasing rumination? Time, frequency? Please add more details. 

Answer: we corrected to - “This study suggests that heat stress has a negative impact on dairy cows, resulting in considerable decreases in their eating and drinking habits, reduction in rumination time, and alterations in their locomotion patterns as well as in their blood gas parameters”  

Line 28 Please rephrase this sentence: …in a Lithuanian dairy farm with 850 milking cows. 

Answer: we corrected to - “This study was conducted during the summer, from June 15 to July 8, 2023, on a Lithuanian dairy farm with 850 milking cows”  

Line 33 The date should be corrected: (period: 2023.07.01. – 2023.07.08.)  

Answer: corrected to - “..(period: 2023.07.01. – 2023.07.08)…" 

Line 33 Please harmonize within the manuscript: “Rumiwatch” or “ RumiWatch”? 

Answer: corrected to - “RumiAatch” 

Line 38 How can the parameters be measured once a week if the entire period only lasted 5 days? Wouldn't it have made more sense to examine several blood samples in the different collection periods? 

Answer: We corrected to - “Blood gas parameters were registered from 2023.06.15 until 2023.07.30, once per week” 

Line 40 Are the differences between the weeks expressed as percentages or percentage points? 

Answer: The differences are expressed as Mean percentage difference. 

Line 44 What activities are meant here? Please provide more details regarding the definition. 

Answer:  

Line 45 This sentence is too long and, in my view, contains some duplications. Please shorten the sentence and focus on the most important facts. 

Answer:  

Line 51 The abstract does not contain any results on blood gas values. So why do you refer to it? 

Answer: We added information - “We found high statistically significant mean differences in the blood urea nitrogen (BUN) levels during the weeks with higher THI (>78) values compared to weeks with THI levels of 72–78 (P<0.001)”  

Line 53 The term global warming is too general. Use another term that describes “heat stress”. As already mentioned the methods used are not very innovative. Please use another term. Please add the terms “blood gas parameters” and “behavior parameters” within the key words. 

Answer: we corrected to - “thermal stress; blood gas parameters; behavior parameters; dairy cattle”  

Line 55 General comment: The introduction should be rewritten, as it is very unstructured and does not lead the reader to the topic as intended. I would suggest starting with a short section on climate change and the expected consequences for the dairy industry. Then it would be useful to describe some of the effects of heat stress on the behavior and health of cows (e.g. feed intake, water intake, rumination, activities). Finally, the various methods that can be used to identify these changes (e.g. sensors, blood gas parameters) could be presented.  

Answer: Following the recommendations, we began with a concise section on climate change and its projected impacts on the dairy sector. We then detailed the effects of heat stress on cows' behaviour and health, including aspects like feed and water consumption, rumination, and their activities. To conclude, we introduced a range of techniques for detecting these changes, such as utilizing sensors and analysing blood gas parameters. 

Line 57 The results of this study should not be cited in the introduction as justification for the usefulness of the study. The research gap should be identified on the basis of a comprehensive literature review. 

Answer: we corrected whole introduction section.  

Line 77 The sentence is too long. Please split into two sentences. 

Answer: Corrected  

 Line 78 Please insert the term “they are” between “when” and “outside”. 

Answer: corrected to - “...when they are outside...”  

 Line 83 The abbreviation “DMI” was not introduced before. Please write out the full term the first time it is mentioned 

Answer: “... dry matter intake (DMI)..”  

 Line 105 Please use the term “Precision livestock farming”, because the following explanations are not limited to cows. 

Answer: corrected to - “Precision livestock farming” in whole manuscript. 

Line 123 Are there really so few studies on the subject? Research has been going on for many years on the subject of "heat stress" and some groups will certainly have used this type of sensor. Please cite some original papers to specifically identify the research gap. 

Answer: we corrected and added references - “To our knowledge, there is not a considerable number of studies that exist, which evaluate the connection between HS and RWS-registered biomarkers [13], [14], [8], [15]”  

Line 131 Comments on the need for further research actually belong in the conclusions and not in the introduction. Please move this section to the other chapter. 

Answer:  we deleted this sentence.  

Line 142 The abbreviations in the following sentences were not introduced before. Please write out the full term the first time it is mentioned. 

Answer: we rewrote the whole introduction section without these abbreviations.  

Line 157 The objective of the study remains unclear. Please specify the objectives of the study and emphasize the novelty value. 

Answer: we corrected to - “This study hypothesizes that heat stress adversely affects dairy cows rumination, eating and drinking behaviors, changes in their locomotor parameters registered by innovative technologies and significant variations in their blood gas parameters. The aim of this study is to investigate the impacts of different risks of heat stress on dairy cows rumination, eating and drinking behaviors, changes in their locomotor parameters registered by innovative technologies and significant variations in their blood gas parameters.” 

Line 168 Why was the ration calculated for a 500 kilogram cow when the herd weighs 600 kilograms on average?  

Answer: this was a mistake, we corrected this sentence to - “This specific diet plan was developed specifically to supply enough nutrition to a 600 kg Holstein cow that produced 37 kg of milk every day”   

Line 180 Why were only 350 cows examined and primiparous cows not included? 

Answer: We added information - “In the current study, 350 fresh (from first until 60 days post partum), multiparous (second and more lactation)...” and corrected title - “The Impacts of  Heat Stress on Rumination, Drinking, and Locomotory Behavior, Registered by Innovative Technologies, and Blood Gas Parameters In Fresh Multiparous Dairy Cows” and aim - “The aim of this study was to investigate the impacts of heat stress on rumination, drinking, and locomotory behavior, registered by innovative technologies, and blood gas parameters in fresh multiparous dairy cows. 

Line 183 Please justify the selection of this THI formula and the associated threshold values. There are different views on the use of the respective formulas in the scientific debate. 

Answer: we corrected to - “The formula used to calculate the temperature humidity index was THI = 0.8 × T + RH × (T 14.4) + 46.4 [21]”  

Line 186 Please insert the term “groups” between “3” and “[20]”. 

Answer: we corrected to - “..into 3 groups [21]:..” 

Line 186 23.06.2023 is included in the first two survey periods. Why have you not made a clear separation of the data sets here? 

Answer: we corrected to - “I. high risk of HS—THI > 78 (period: 2023.06.15- 06.23); II. medium risk of HS—THI 72-78 (period: 2023.06.24 – 06.30); and III. low risk pf HS—THI < 72 (period: 2023.07.01- 08)” 

Line 199 Please give an example of “other jaw movements”. 

Answer:  we corrected to - “other chewing activities (n/h)—number of jaw movements performed during the chosen summary interval”  

Line 214 Maybe “rumination, feeding or drinking activity”? 

Answer: we corrected to - “other activities = activities not classified as any rumination, feeding or drinking activity’  

Line 235 Please harmonize within the manuscript “p” or “P”; and “p<0.05” or “p < 0.05”! 

Answer:  we corrected to “P” in whole manuscript.  

Line 238 There is no reference to the presentation of the data in Table 1. 

Answer: we added information - “The data analysis of the RWS parameters presented the highest statistically significant mean differences between the investigated indicators during different risk scenarios of HS (Table 1)"  

Line 245 Please delete the second bracket after “P<0.001”. 

Answer: deleted.  

Table 1 Please widen the table to make the contents easier to read. Tables should be self-explanatory, so abbreviations should be written out in full or integrated into footnotes. Use the same number of decimal places in the table. Why do you use the superscript d in addition to a,b,c? What is the significance of this? Why is there an asterisk in the line "other chew"? I would sort the rows according to the individual behaviors (rumination, eating, activity, etc.). This makes it easier to compare the results with each other. Why is the p-value written in italics? 

Answer: We corrected Table 1 accordin to yours suggestions.  

Line 255 There is no reference to the presentation of the data in Figure 1. Please add the units of the values. 

Answer: We corrected to - “The mean values for the eating time changes assessed every 24 h showed that they changed from 4.21 to 13.61 min/h ; the drinking time changed from 0.20 to 0.52 min/h chews per bolus changed from 2.91 to 7.94 n/min; up time changed from 4.24 to 17.03 min/h; other activity changed from 16.56 to 27.19; other chew changed from 87.77 to 173.72 min/h; chews per minute changed from 50.77 to 73.41n/min; activity changed from 60.99 to 121.82 min/h; rumination time changed from 15.19 to 29.9 min/h; bolus changed from 16.4 to 31.64 n/h; and down time changed from 16.48 to 24.95 min/h (Figure 1A, B and C)”  

Figure 1 The units on the y-axis are missing. Changes in the behavior of cows during the course of the day have been known for a long time and are mainly influenced by milking times and feeding. Wouldn't it make more sense here to limit ourselves to a few parameters and compare these between the survey periods? For example, how does feeding behavior change during heat stress compared to cooler periods? Do the animals eat more during the cooler night and should the feeding times be adjusted accordingly? 

Answer: Thank You very much for Your suggestion, we have changed the Figure 1, also have added the Table 3 to show the relationship of HS group (groups of THI) with investigated parameters. Also, we can see from the Table 1, when THI is the highest drinking time is higher, rumination time, eat down time is lower, while eat up time (in the morning not o hot) is a little bit higher. 

Line 271 The abbreviation “BUN” was not introduced before. Please write out the full term the first time it is mentioned 

Answer: we corrected to - “... blood urea nitrogen (BUN)…" 

Line 275 Please replace the comma through a full stop (16.89). 

Answer: Corrected to - “... 16.89...” 

Table 2 Tables should be self-explanatory, so abbreviations should be written out in full or integrated into footnotes. Why do you use the superscript d in addition to a,b,c? What is the significance of this? Why is the p-value written in italics? 

Answer: we added information - “pH—hydrogen potential; pCO2—partial carbon dioxide pressure; pO2—partial oxygen pressure; HCO3—bicarbonate; BE (ecf)- BE (efc)—base excess in the extracellular fluid; sO2—oxygen saturation; Na+—sodium; K+—potassium; Ca++—ionized calcium; Cl-chlorides; TCO2—total carbon dioxide in the blood; Hct—hematocrit; cHgb - hemoglobin concentration; BE—base excess in the blood; Glu—glucose; Lac—lactate; BUN -blood urea nitrogen; Crea- creatinine; BUN/Crea – blood urea nitrogen and creatinine ratio”  We deleted “d” and corrected the p – value.  

Line 284 Please insert “gas parameters” after the term “RWS parameters and blood” 

Answer: we corrected to - “..RWS and blood gas parameters..”   

Line 285 This is already part of the discussion and should not be mentioned in the results chapter. 

Answer: we removed this information to the discussion part. 

Table 3 Please widen the table to make the contents easier to read. Tables should be self-explanatory, so abbreviations should be written out in full or integrated into footnotes.  

Answer: we corrected this table and added information - “pH—hydrogen potential; pCO2—partial carbon dioxide pressure; pO2—partial oxygen pressure; HCO3—bicarbonate; BE (ecf)- BE (efc)—base excess in the extracellular fluid; sO2—oxygen saturation; Na+—sodium; K+—potassium; Ca++—ionized calcium; Cl-chlorides; TCO2—total carbon dioxide in the blood; Hct—hematocrit; cHgb - hemoglobin concentration; BE—base excess in the blood; Glu—glucose; Lac—lactate; BUN -blood urea nitrogen; Crea- creatinine; BUN/Crea – blood urea nitrogen and creatinine ratio”  

Line 297 Why is it an attempt of the industry? I think it is a natural bevioral change of HS cows? 

Answer: we corrected to - “Decreased feed intake is a conservative response among HS animals”  

Line 205 Please harmonize within the manuscript “25 °C” or “25°C”. 

Answer: corrected to - “25 °C”  

Line 308 The abbreviation HS was introduced before and must not be mentioned again 
Answer: we corrected to - “..different HS risk...” 

Line 311 Please cite some studies using also THI thresholds instead of ° Celsius in order to compare the results. Alternatively you can add some more details concerning temperature and humidity in the Material and Method chapter. 

Answer: We added information in discussion section - “The findings of other studies indicate a strong inverse relationship (r = -0.82) between THI and DMI, with an increase in THI of one unit resulting in a daily reduction of DMI of 0.45 kg [28]. Furthermore, by controlling metabolic rate and heat production, a change in DMI served as a protective mechanism to preserve thermal balance [29]. When dairy cows were exposed to cold conditions (THI = 38.9) as opposed to hot conditions (THI = 73.9), Hill and Wall (2017) [30] found that there was an 11.5% increase in DMI. In contrast, our findings were corroborated by the finding that DMI decreased by 0.51 kg for each unit rise in THI between 73 and 83 units [23]. Specifically, inconsistent DMI responses to THI conditions and milk output are probably the result of disturbed energy metabolism under ongoing cold-wet conditions [31].” 

Also, in materials and methods section we added - “The formula used to calculate the temperature humidity index was THI = 0.8 × T + RH × (T 14.4) + 46.4 [21]”  

Line 316 The abbreviation “TNZ” was not introduced before. Please write out the full term the first time it is mentioned 

Answer: we corrected to - “In comparison to cows in the thermos comfort zone... “  

Line 321 Why did you use RWS as an abbreviation for real-time welfare scoring? 

Answer: we corrected to - “The current study supports these observations, especially in relation to real-time welfare scoring parameters”  

Line 325 However, in your study there is no linear relationship between heat stress and walking time! 

Answer:  We deleted this statment  

Line 344 Please insert a paragraph here. 

Answer: Corrected  

Line 344 This section does not promote knowledge gain. I would again suggest to focus on behavioral alterations between the three groups. 

Answer: we corrected whole discussion section.  

Line 371 Please insert a reference for this statement. 

Answer: we deleted this statement 

Line 373 Please discuss in more detail why there are no connections between the RWS indicators and the blood gas parameters. Is it because of the methodology or do you generally not suspect a connection? What does this lack of connection mean for the usefulness of the RWS indicators? 

Answer: The lack of connection between RWS indicators and blood gas parameters in our study, which aimed to investigate the effects of heat stress on these factors in fresh multiparous dairy cows, could be due to methodological differences or an inherent lack of direct correlation between behavioral and physiological responses. Despite this, the RWS indicators remain valuable for monitoring behavioral responses to heat stress and informing farm management practices. This suggests that while these indicators and blood parameters provide different insights, both are crucial for a comprehensive understanding of cows' responses to heat stress. 

We added information in discussion section - “The study presented some limitations, including its single-farm focus on Holstein cows during a specific period, limiting its generalizability. Also, the analysis of the investigated parameter relationships reveals a weak correlation between rumination, drinking, and locomotor behavior, and blood gas parameters, with the statistically significant correlation coefficients ranging from 0.323 to 0.375. This is likely due to the small sample size of animals used in the study. Factors, such as individual variability, external influences on behavior, and the absence of long-term impact assessments, highlight the need for further research to refine heat stress management strategies in dairy farming practices. Despite these limitations, the study provided valuable insights into the dynamic nature of ruminant behavior and blood parameters resulting from heat stress, laying the groundwork for future research on this area and improved management practices”  

Line 405 Please go into detail about the practical benefits of the work. How can farmers benefit from your insights? 

Answer: we added in conclusions section - “Based on our study's findings, we advise dairy farmers to employ innovative technologies for the real-time monitoring and management of heat stress in cows. During periods of high THI (> 78), it is crucial to closely monitor changes in activity time and chewing behaviors, as significant increases in these behaviors are indicative of elevated heat stress levels. Farmers should also adjust their nutritional plans based on continuous monitoring of blood urea nitrogen levels. This innovative approach, leveraging advanced technologies, is vital for maintaining the health and productivity of dairy cows under various climatic conditions. 

Line 415 Please do not use the term “robust statistical analysis” if you rightly pointed out the small sample size beforehand 

Answer: corrected.  

Line 445 The name of the first author is not correct. 

Answer:  

Line 497 Maybe “Animal Science Papers” instead of “Animal science papers”? 

Answer: we corrected to - “Animal Science Papers and Reports”  

Line 521 Please delete “1” after the title. 

Answer: Corrected  

Line 525 The names of the authors are not correct. 

Answer: corrected. 

Line 533 Maybe “Animal” instead of “animal”? 

Answer: Corrected  

Line 535 Please delete “2” after the title. 

Answer: Corrected  

Line 541 Please delete “2,3” after the title. 

Answer: Deleted  

Reviewer 3 Report

Comments and Suggestions for Authors

The manuscript is focused on the study of various behavioral parameters of dairy cows, as well as blood parameters and how they are affected by heat stress. The main strength of the manuscript lies in the large number of cows used, although as a weakness it can be mentioned that the experiment was carried out only once and on a single farm. This fact is discussed in the manuscript itself.

In my opinion, the introduction could be improved because it is very long but it does not adequately focus the problems and the objective of the work. In addition, lines 57 to 63 include a paragraph that corresponds to results and not to discussion.

Conclusion statement did not satisfy the hipothesis of word detailed in L154-156. That statement is too vague and do not provide information. 

The material and methods section is somewhat confusing and should be improved. sections 2.1 and 2.2 could be put together and summarized. The fact that there are 850 cows on the farm does not add anything, if 350 of them have been studied. It is important to clarify how the analysis of variance of the THI has been done: how many observations are in each group, what distribution, and more information on this should be provided since the study is based on these data.

Expressions such as "RWS indicators" or "RWS parameters" are frequently used. I suggest eliminating all these and talking about the variables studied. Once it has been explained with which device they have been measured, the important thing is the variable and not what it has been measured with.

Table 1. I do not understand why regressions are used to study these variables. It would be better use an ANOVA in the same way that Table 2. to improve clarity of  both tables, I suggest to delete the error of the mean and use other kind of error such as s.e. of the model or RMSE or RSE, and level of signification.

L253 are the differences between weeks or between HS groups? Change it.

Table 3. I suggest show only significant correlations.

Figure 1. Figure 1, as is, do not provide any information about the heat stress which is the main interest of manuscript. In my opinion, this Figure 1 is not useful to accomplish the aim of the manuscript. Maybe you want to study how heat stress modify daily patterns of the studied variables. In this case, you should study interactions between HS and time for each variable.

Author Response

Dear Reviewer,  

Authors are very thankful for the comments, which help us to improve the manuscript. All changes proposed have been included in the manuscript and highlighted in yellow and track changes.   

Best Regards,  

Prof. Ramunas Antanaitis 

You can find your questions and comments along with our responses here: 

In my opinion, the introduction could be improved because it is very long but it does not adequately focus the problems and the objective of the work. In addition, lines 57 to 63 include a paragraph that corresponds to results and not to discussion. 

Answer: we corrected the introduction section following- we began with a concise section on climate change and its projected impacts on the dairy sector. We then detailed the effects of heat stress on cows' behavior and health, including aspects like feed and water consumption, rumination, and their activities. To conclude, we introduced a range of techniques for detecting these changes, such as utilizing sensors and analyzing blood gas parameters. 

Conclusion statement did not satisfy the hipothesis of word detailed in L154-156. That statement is too vague and do not provide information.  

Answer: we corrected aim of our study- “. This study hypothesizes that heat stress adversely affects dairy cows rumination, eating and drinking behaviors, changes in their locomotor parameters registered by innovative technologies and significant variations in their blood gas parameters. The aim of this study was to investigate the impacts of heat stress on rumination, drinking, and locomotory behavior, registered by innovative technologies, and blood gas parameters in fresh multiparous dairy cows”  

Also, conclusion section following- “We can conclude that heat stress significantly impacts activity time, chewing behaviors, and general activity, as registered by innovative technologies, as well as blood gas parameters in fresh multiparous dairy cows. We found that specifically during the high heat stress (HS) risk period (THI > 78), there was an 11.75% increase in other activity time and a 17.67% increase in other chewing behaviors compared to the low-risk period (THI < 72). Even in the medium HS risk period (THI 72–78), these behaviors were elevated (13.80% increase in chewing behaviors) compared to low-risk conditions. Interestingly, a 12.82% increase in activity change was observed when transitioning from medium to low HS risk. We observed a substantial and statistically significant increase in BUN levels during weeks with higher THI values (>78) compared to those with moderate THI levels (72–78). Specifically, BUN levels were 19.75% higher in weeks where THI exceeded 78, compared to weeks with THI values between 72 and 78. This increase is noteworthy as it was 16.89% higher than the urea level for the same period. Moreover, when comparing the highest THI values (>78) to the second and third weeks, urea levels were found to be 21.16% and 18.81% higher, respectively.  

Based on our study's findings, we advise dairy farmers to employ innovative technologies for the real-time monitoring and management of heat stress in cows. During periods of high THI (> 78), it is crucial to closely monitor changes in activity time and chewing behaviors, as significant increases in these behaviors are indicative of elevated heat stress levels. Farmers should also adjust their nutritional plans based on continuous monitoring of blood urea nitrogen levels. This innovative approach, leveraging advanced technologies, is vital for maintaining the health and productivity of dairy cows under various climatic conditions. 

The material and methods section is somewhat confusing and should be improved. sections 2.1 and 2.2 could be put together and summarized.  

Answer: we did one section - “2.1. Farm, experiment design and duration”  

The fact that there are 850 cows on the farm does not add anything, if 350 of them have been studied. It is important to clarify how the analysis of variance of the THI has been done: how many observations are in each group, what distribution, and more information on this should be provided since the study is based on these data. 

Answer: We corrected to - “This study was performed during the summer period (2023.06.15–2023.07.08) on one Lithuanian dairy farm. There were 850 milking cows on the farm. In the current study, 350 fresh (from first until 60 days post partum), multiparous (second and more lactation)..”  

Also, according to this we corrected title and aim of our study.  

Expressions such as "RWS indicators" or "RWS parameters" are frequently used. I suggest eliminating all these and talking about the variables studied. Once it has been explained with which device they have been measured, the important thing is the variable and not what it has been measured with. 

Answer: we changed “RWS” with “rumination, drinking, and locomotory behavior parameters”  

Table 1. I do not understand why regressions are used to study these variables. It would be better use an ANOVA in the same way that Table 2. to improve clarity of  both tables, I suggest to delete the error of the mean and use other kind of error such as s.e. of the model or RMSE or RSE, and level of signification. 

L253 are the differences between weeks or between HS groups? Change it. 

Answer: we corrected - “The letters a, b, and c indicate the statistically significant differences between HS groups; P < 0.05”. 

Table 3. I suggest show only significant correlations. 

Figure 1. Figure 1, as is, do not provide any information about the heat stress which is the main interest of manuscript. In my opinion, this Figure 1 is not useful to accomplish the aim of the manuscript. Maybe you want to study how heat stress modify daily patterns of the studied variables. In this case, you should study interactions between HS and time for each variable. 

Answer: We have made corrections (Figure 1); the new figures are added according to THI and the change of investigated traits per day (24 h). 

Reviewer 4 Report

Comments and Suggestions for Authors

This study used the RumiWatch sensor system and blood gas parameters to investigate the behavior and metabolism of dairy cows under different THI conditions.

Unfortunately, several key aspects are unclear in the manuscript and the analysis has multiple flaws which limits the ability to determine if the findings and conclusions are robust.

One main issue is the confusion around the periods used and animals present during these periods:

The authors state multiple times that 3 periods were considered, corresponding to high, medium, and low heat stress risk. However, it is not clear how long these periods were as the dates are not consistent across the paper (data used during the acclimatization period or not?). Also, it is not clear if all 350 cows were followed through these 3 periods or if the group changed. It is also not indicated how the cows were housed (group size, group composition change etc), which may impact what the experimental unit is.
Importantly, it is not clear if any of the periods were without heat stress? This way the different levels of heat stress might be compared to each other but we don’t get an idea about the deviation from normal.

The other main concern is that throughout the paper multiple analysis is presented to answer the same question, basically the same results are indicated as text, table, figure, anova, correlation… I would suggest to the authors to clean this up, choose the most appropriate analysis (which will depend on the experimental unit and the distribution of cows among periods which we don’t know at this point).
Using correlations when heat stress risk has only been assessed in 3 categories is not appropriate. Present the results only in the results section and only interpret these in the discussion. No need to repeat results numerically in the discussion and conclusions.

Detailed comments:

L21: Decrease in drinking? I suggest saying changes in drinking behavior

L32: Make sure this is the same number as in the text - there you say 11,400kg

L38: The last period ended on July 8 as mentioned above, please clarify

L39-40: But above you say that the high HS risk period started on 06.15 already and the low HS risk period ended only 07.08? Please clarify the dates, measuring only between June 20 and July 03 contradicts what is said above

L42-43: Were the 3 THI periods a finding or a characteristic of the data that you used for analysis? Anyways, this is already said above, please combine and clarify

L45: Here and throughout the paper: No need to restate the THI every time. Define the 3 periods and the corresponding dates and THI and then refer to the periods only. I suggest using “low HS”, “medium HS” and “high HS” period as names, then readers know what you refer to. Use one term consistently throughout the analysis and the manuscript text. Having “period1-3”, “THI 1-2-3”, “HS risk”, “HS period”… is very confusing.

L134: How was water provided? Drinker type and number of drinkers?

L148-151: Start section 2.1 with this

L156-158: How were the cows housed? What was the group size? Did cows differ between periods or were the same 350 cows followed for 3 weeks?

L165-184: It is very hard to follow the variables like this, please put in a table form!

L201-204: Define periods at the beginning as mentioned above

L210-211: This was already stated at the beginning, please only define once

L225-246: Why does it make sense to look at each hour separately? Did you have a corresponding hypothesis? This is a lot of comparisons… Have you corrected for multiple testing?

Figure 1: Why do the plots have bar charts for the first period and lines for the second two?

Table 2: Use the same name, not HS group. The 3 different heat stress risk periods have been used as a categorical variable, correlation is not appropriate here as only 3 categories were considered.

Table 4: No point in correlating all variables with each other without a specific hypothesis. Also, this needs to be corrected for multiple testing.

Due to the concerns with the analysis and results I did not consider the discussion in detail.

Comments on the Quality of English Language

Please send the manuscript for English language editing to correct grammar and improve the understandability and flow of the paper.